# SpikeBERT: A Language Spikformer Learned from BERT with Knowledge Distillation

## Abstract

Spiking neural networks (SNNs) offer a promising avenue to implement deep neural networks in a more energy-efficient way. However, the network architectures of existing SNNs for language tasks are still simplistic and relatively shallow, and deep architectures have not been fully explored, resulting in a significant performance gap compared to mainstream transformer-based networks such as BERT. To this end, we improve a recently-proposed spiking Transformer (i.e., Spikformer) to make it possible to process language tasks and propose a two-stage knowledge distillation method for training it, which combines pre-training by distilling knowledge from BERT with a large collection of unlabelled texts and fine-tuning with task-specific instances via knowledge distillation again from the BERT fine-tuned on the same training examples. Through extensive experimentation, we show that the models trained with our method, named SpikeBERT, outperform state-of-the-art SNNs and even achieve comparable results to BERTs on text classification tasks for both English and Chinese with much less energy consumption.

## 1 Introduction

Modern artificial neural networks (ANNs) have been highly successful in a wide range of natural language processing (NLP) and computer vision (CV) tasks. However, it requires too much computational energy to train and deploy state-of-the-art ANN models, leading to a consistent increase of energy consumption per model over the past decade. The energy consumption of large language models during inference, such as ChatGPT (OpenAI, 2022) and GPT-4 (OpenAI, 2023), is unfathomable. In recent years, spiking neural networks (SNNs), arguably known as the third generation of neural network (Maas, 1997), have attracted a lot of attention due to their high biological plausibility, event-driven property and low energy consumption (Roy et al., 2019). Like biological neurons, SNNs use discrete spikes to process and transmit information. Nowadays, neuromorphic hardware can be used to fulfill spike-based computing, which provides a promising way to implement artificial intelligence with much lower energy consumption.

Spiking neural networks have achieved great success in image classification task (Hu et al., 2018; Yin et al., 2020; Fang et al., 2021; Ding et al., 2021; Kim et al., 2022b; Zhou et al., 2022) and there have been some studies (Plank et al., 2021; Lv et al., 2023; Zhu et al., 2023) that have demonstrated the efficacy of SNNs in language tasks. However, the backbone networks employed in SNNs for language tasks are relatively simplistic, which significantly lower the upper bound on the performance of their models. For instance, the SNN proposed by Lv et al. (2023), which is built upon TextCNN (Kim, 2014), demonstrates a notable performance gap compared to those built on Transfomer-based (Vaswani et al., 2017) language models like BERT (Devlin et al., 2019) and RoBERTa (Liu et al., 2019) on multiple text classification benchmarks.

Recently, Spikformer has been proposed by Zhou et al. (2022), which firstly introduced Transformer architecture to SNNs and significantly narrowed the gap between SNNs and ViT (Dosovitskiy et al., 2020) on ImageNet (Deng et al., 2009) and CIFAR-10. In this study, we aim to explore the feasibility of applying deep architectures like Spikformer to the natural language processing field and investigate how far we can go with such deep neural models on various language tasks. As shown in Figure 1, considering the discrete nature of textual data, we improve the architecture of Spikformer to make it suitable for language tasks. we replace certain modules that were originally designed for image processing with language-friendly modules (see Section 3.2 for details on the improvement

Figure 1: (a) Architecture of Spikformer (Zhou et al., 2022) with $L$ encoder blocks. Spikformer is specially designed for image classification task, where spiking patch splitting (SPS) module and "convolution layer + batch normalization" module can process vision signals well. And the spiking self attention (SSA) module in Spikformer aims to model the attention between every two dimensions so that we denote it as "D-SSA". (b) Architecture of SpikeBERT with $L'$ encoder blocks. In order to improve the model's ability of processing texts, we adopt "linear layer + layer normalization", and also replace the SPS module with a word embedding layer. Furthermore, we modify the SSA module to enhance SpikeBERT's ability to concentrate on the interrelation between all pairs of words (or tokens), instead of dimensions.

in network architecture). In general, a deeper ANN model can often bring better performance, and increasing the depth of an ANN allows for the extraction of more complex and abstract features from the input data. However, Fang et al. (2020b) have shown that deep SNNs directly trained with backpropagation through time (Werbos, 1990) using surrogate gradients (See Section2.1) would suffer from the problem of gradient vanishing or exploding due to "self-accumulating dynamics". Therefore, we propose to use knowledge distillation (Hinton et al., 2015) for training language Spikformer so that the deviation of surrogate gradients in spiking neural networks will not be rapidly accumulated. Nevertheless, how to distill knowledge from ANNs to SNNs is a great challenge because the features in ANNs are floating-point format while those in SNNs are spike trains and even involve an additional dimension T (time step). We find that this problem can be solved by introducing external modules for aligning the features when training (See Section 3.3.1).

Inspired by the widely-used "pre-training + fine-tuning" recipe (Sun et al., 2019; Liu, 2019; Gururan-gan et al., 2020), we present a two-stage knowledge distillation strategy. In the first stage, we choose the representative example of large pre-trained models, BERT, as the teacher model and SpikeBERT as the student model. We utilize a large collection of unlabelled texts to align features produced by two models in the embedding layer and hidden layers. In the second stage, we take a BERT fine-tuned on a task-specific dataset as a teacher and the model after the first pre-training stage as a student. We first augment the training data for task-specific datasets and then employ the logits predicted by the teacher model to further guide the student model. After two-stage knowledge distillation, a spiking language model, named SpikeBERT, can be built by distilling knowledge from BERT.

The experiment results show that SpikeBERT can not only outperform the state-of-the-art SNNs-like frameworks in text classification tasks but also achieve competitive performance to BERTs. In addition, we calculate the theoretical energy consumption of running our SpikeBERT on a 45nm neuromorphic hardware (Horowitz, 2014) and find that SpikeBERT demands only about 27.82% of the energy that fine-tuned BERT needs to achieve comparable performance. The experiments of the ablation study (Section 4.5) also show that "pre-training distillation" plays an important role in training SpikeBERT.

In conclusion, the major contribution of this study can be summarized as follows:

- We improve the architecture of Spikformer for language processing and propose a two-stage, "pre-training + task-specific" knowledge distillation training method, in which SpikeBERTs are pre-trained on a huge collection of unlabelled texts before they are further fine-tuned on task-specific datasets by distilling the knowledge of feature extractions and predictive power from BERTs.
- We empirically show that SpikeBERT achieved significantly higher performance than existing SNNs on 6 different language benchmark datasets for both English and Chinese.
- This study is among the first to show the feasibility of transferring the knowledge of BERT-like large language models to spiking-based architectures that can achieve comparable results but with much less energy consumption.

## 2 RELATED WORK

### 2.1 SPIKING NEURAL NETWORKS

Different from traditional artificial neural networks, spiking neural networks utilize discrete spike trains instead of continuous decimal values to compute and transmit information. Spiking neurons, such as Izhikevich neuron (Izhikevich, 2003) and Leaky Integrate-and-Fire (LIF) neuron (Wu et al., 2017), are usually applied to generate spike trains. However, due to the non-differentiability of spikes, training SNNs has been a great challenge for the past two decades. Nowadays, there are two mainstream approaches to address this problem.

**ANN-to-SNN Conversion** ANN-to-SNN conversion method (Diehl et al., 2015; Cao et al., 2015; Rueckauer et al., 2017; Hu et al., 2018) aims to convert weights of a well-trained ANN to its SNN counterpart by replacing the activation function with spiking neuron layers and adding scaling rules such as weight normalization (Diehl et al., 2016) and threshold constraints (Hu et al., 2018). This approach suffers from a large number of time steps during the conversion.

**Backpropagation with Surrogate Gradients** Another popular approach is to introduce surrogate gradients (Neftci et al., 2019) during error backpropagation, enabling the entire procedure to be differentiable. Multiple surrogate gradient functions have been proposed, including the Sigmoid surrogate function (Zenke & Ganguli, 2017), Fast-Sigmoid (Zheng & Mazumder, 2018), ATan (Fang et al., 2020b), etc. Backpropagation through time (BPTT) (Werbos, 1990) is one of the most popular methods for directly training SNNs(Shrestha & Orchard, 2018; Kang et al., 2022), which applies the traditional backpropagation algorithm (LeCun et al., 1989) to the unrolled computational graph. In recent years, several BPTT-like training strategies have been proposed, including SpatioTemporal Backpropagation (STBP) (Wu et al., 2017), STBP with Temporal Dependent Batch Normalization (STBP-tdBN) (Zheng et al., 2020), and Spatio-Temporal Dropout Backpropagation (STDB) (Rathi et al., 2020). These strategies have demonstrated high performance under specific settings. We choose BPTT with surrogate gradients as our training method. For more detailed information about Backpropagation Through Time (BPTT), please refer to Appendix A.

### 2.2 KNOWLEDGE DISTILLATION

Hinton et al. (2015) proposed the concept of knowledge distillation by utilizing the "response-based" knowledge (i.e., soft labels) of the teacher model to transfer knowledge. However, when this concept was first proposed, the features captured in the hidden layers were neglected, as they only focused on the final probability distribution at that time. To learn from teacher models more efficiently, some works (Zagoruyko & Komodakis, 2016; Heo et al., 2019; Chen et al., 2021) have advocated for incorporating hidden feature alignment during the distillation process. In addition, relation-based knowledge distillation has been introduced by Park et al. (2019), demonstrating that the interrelations between training data examples were also essential.

Recently, there have been a few studies (Kushawaha et al., 2020; Takuya et al., 2021; Qiu et al., 2023) in which knowledge distillation approaches were introduced to train SNNs. However, most of them focused on image classification tasks only, which cannot be trivially applied to language tasks. In this study, we propose a two-stage knowledge distillation approach to train the proposed SpikeBERT for text classification tasks, which is among the first ones to show the feasibility of transferring the knowledge to SNNs from large language models.

## 3 METHOD

In this section, we describe how we improve the architecture of Spikformer and introduce our two-stage distillation approach for training SpikeBERT. Firstly, we will depict how spiking neurons and surrogate gradients work in spiking neural networks. Then we will show the simple but effective modification of Spikformer to enable it to represent text information. Lastly, we will illustrate "pre-training + task-specific" distillation in detail.

### 3.1 Spiking Neurons and Surrogate Gradients

Leaky integrate-and-fire (LIF) neuron (Wu et al., 2017) is one of the most widely used spiking neurons. Similar to the traditional activation function such as ReLU, LIF neurons operate on a weighted sum of inputs, which contributes to the membrane potential $U_t$ of the neuron at time step $t$. If membrane potential of the neuron reaches a threshold $U_{\text{thr}}$, a spike $S_t$ will be generated:

$$S_t = \begin{cases} 1, & \text{if } U_t \geq U_{\text{thr}}; \\ 0, & \text{if } U_t < U_{\text{thr}}. \end{cases} \tag{1}$$

We can regard the dynamics of the neuron's membrane potential as a resistor-capacitor circuit (Maas, 1997). The approximate solution to the differential equation of this circuit can be represented as follows:

$$U_t = I_t + \beta U_{t-1} - S_{t-1} U_{\text{thr}}, \quad I_t = W X_t \tag{2}$$

where $X_t$ are inputs to the LIF neuron at time step $t$, $W$ is a set of learnable weights used to integrate different inputs, $I_t$ is the weighted sum of inputs, $\beta$ is the decay rate of membrane potential, and $U_{t-1}$ is the membrane potential at time $t-1$. The last term of $S_{t-1}U_{\text{thr}}$ is introduced to model the spiking and membrane potential reset mechanism.

In addition, we follow Fang et al. (2020a) and use the Arctangent-like surrogate gradients function, which regards the Heaviside step function (Equation 1) as:

$$S \approx \frac{1}{\pi} \arctan(\frac{\pi}{2}\alpha U) + \frac{1}{2} \tag{3}$$

Therefore, the gradients of $S$ in Equation 3 are:

$$\frac{\partial S}{\partial U} = \frac{\alpha}{2} \frac{1}{(1 + (\frac{\pi}{2}\alpha U)^2)} \tag{4}$$

where $\alpha$ defaults to 2.

### 3.2 The Architecture of SpikeBERT

SpikeBERT is among the first large spiking neural network for language tasks. Our architecture is based on Spikformer (Zhou et al., 2022), which is a hardware-friendly Transformer-based spiking neural network and we have shown it in Figure 1 (a). In Spikformer, the vital module is the spiking self attention (SSA), which utilizes discrete spikes to approximate the vanilla self-attention mechanism. It can be written as:

$$\text{SSA}(Q_s, K_s, V_s) = \mathcal{SN}(\text{BN}(\text{Linear}(Q_s K_s^T V_s * \tau)))$$
$$Q_s = \mathcal{SN}_Q(\text{BN}(X_s W_Q)), \quad K_s = \mathcal{SN}_K(\text{BN}(X_s W_K)), \quad V_s = \mathcal{SN}_V(\text{BN}(X_s W_V)) \tag{5}$$

where $\mathcal{SN}$ is a spike neuron layer, which can be seen as Heaviside step function like Equation 1, $X_s \in \mathbb{R}^{T \times L \times D}$ is the input of SSA, $T$ is number of time steps, BN is batch normalization, $\tau$ is a scaling factor. Outputs of SSA and $Q_s, K_s, V_s$ are all spike matrices that only contain 0 and 1. $W_Q, W_K, W_V$ and Linear are all learnable decimal parameters. Please note that the shape of attention map in Spikformer, i.e., $Q_s K_s^T$, is $D \times D$, where $D$ is the dimensionality of the hidden layers.

We modify Spikformer so that it can effectively process textual data. Firstly, we replace spiking patch splitting (SPS) module with a word embedding layer and a spiking neuron layer, which is necessary for mapping tokens to tensors. Most importantly, we think that the features shared with words in different positions by attention mechanism are more important than those in different dimensions. Therefore, we reshape the attention map in spiking self-attention (SSA) module to $N \times N$ instead of $D \times D$, where $N$ is the length of input sentences. Lastly, we replace "convolution layers + batch normalization" with "linear layers + layer normalization" because convolution layers are always used to capture the pixel features in images, which is not suitable for language tasks. We show the architecture of our SpikeBERT in Figure 1 (b).

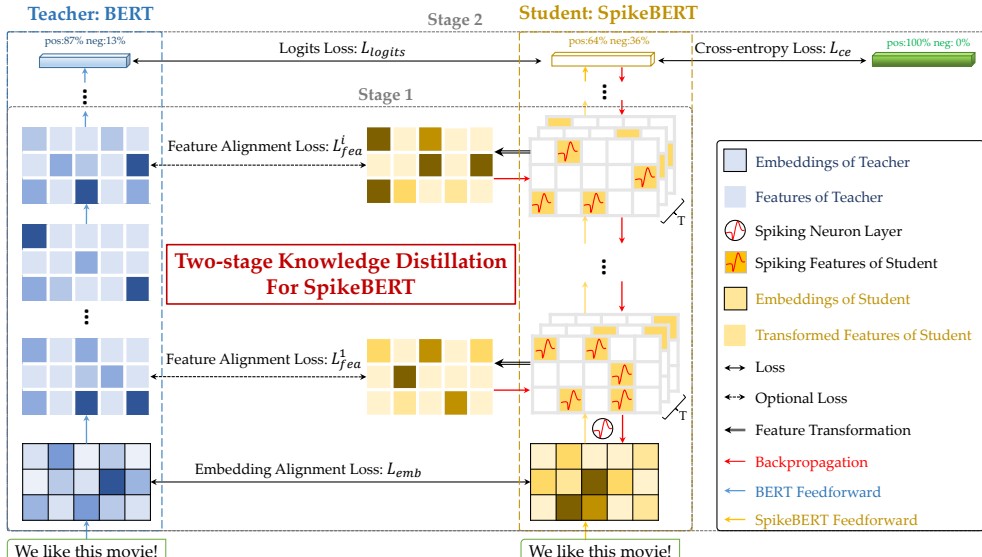

Figure 2: Overview of our two-stage distillation method (pre-training + task-specific distillation) for training SpikeBERT. $T$ is the number of time steps of features in every layer. Notice that the logits loss and cross-entropy loss are only considered in stage 2. The varying shades of color represent the magnitude of the floating-point values. The dotted line under $L_{fea}^i$ indicates that features of some hidden layers can be ignored when calculating feature alignment loss. If the student model contains different numbers of layers from the teacher model, we will align features every few layers.

## 3.3 TWO-STAGE DISTILLATION

We have tried to directly conduct mask language modeling (MLM) and next sentence prediction (NSP) like BERT, but we found that the whole model failed to converge due to "self-accumulating dynamics"(Fang et al., 2020b), which was an unsolved problem in large-scale spiking neural networks. Therefore, we choose to use knowledge distillation for training SpikeBERT so that the deviation of surrogate gradients in the model will not be rapidly accumulated. However, it is a huge challenge to distill knowledge from ANNs to SNNs, for the hidden features are not the same data format in ANNs and SNNs. Features in ANNs are floating-point format while those in SNNs are time-varying spike trains. We find that introducing external modules (See Section 3.3.1) for aligning features when training can perfectly address this problem.

We follow the popular "pre-training + fine-tuning" recipe and propose the two-stage distillation. The first stage is to align the embeddings and hidden features between BERT and SpikeBERT using a large-scale corpus. The second stage is to distill logits and cross-entropy information on a task-specific dataset from a fine-tuned BERT to the model finishing stage 1. We show the overview of our method in Figure 2.

### 3.3.1 THE FIRST STAGE: PRE-TRAINING DISTILLATION

Given a pre-trained BERT (Devlin et al., 2019) irrelevant to downstream tasks as teacher $TM$ and a SpikeBERT as student $SM$, our goal in this stage is to align the embeddings and hidden features of $TM$ and $SM$ with a collection of unlabelled texts. We will introduce embedding alignment loss and feature alignment loss in the following.

**Feature Alignment Loss**  This loss $L_{fea}$ is to measure the similarity of features between $TM$ and $SM$ at every hidden layer. However, the shape of the student model's feature $F_{sm}$ at every layer is $T \times N \times D$ but that of BERT's feature $F_{tm}$ is $N \times D$, where $T$ is the number of time steps, $D$ is the dimensionality of hidden layers and $L$ is sample length. What's more, $F_{sm}$ is a matrix only containing 0 and 1 but $F_{tm}$ is a decimal matrix. To address the issue of different dimensions between $F_{tm}$ and $F_{sm}$, as well as the disparity between continuous features of $TM$ and discrete features of $SM$, a transformation strategy is necessary. We follow the feature transformation approaches of Heo et al. (2019); Chen et al. (2021); Qiu et al. (2023) to map the features of $TM$ and $SM$ to the same

content space:

$$F'_{tm} = F_{tm}, \quad F'_{sm} = \text{LayerNorm}(\text{MLP}(\sum_{t}^{T}(F^t_{sm}))) \tag{6}$$

However, we find it hard to align the features generated by the student model with those generated by BERT for the first few layers in this stage. We think that's because the student model might require more network layers to capture the essential features via the interaction among the inputs. As shown in Figure 2, we choose to ignore some front layers when calculating feature alignment loss. Assume BERT contains $B$ Transformer blocks (i.e., $B$ layers) and assume the student model contains $M$ Spike Transformer Block. Therefore, we will align features every $\lceil \frac{B}{M} \rceil$ layers if $B > M$. For layer $i$ in student model, its feature alignment loss is $L^i_{fea} = ||F'_{tm} - F'_{sm}||_2$.

**Embedding Alignment Loss**  As discussed in Section 3.2, the embeddings of the input sentences are not in the form of spikes until they are fed forward into the Heaviside step function. Define $E_{tm}$ and $E_{sm}$ as the embeddings of teacher and student, respectively so the embedding alignment loss is $L^i_{fea} = ||E_{tm} - \text{MLP}(E_{sm})||_2$. The MLP layer is a transformation playing a similar role as that in Equation 6.

To sum up, in stage 1, the total loss $L_1$ is the sum of the chosen layer's feature alignment loss:

$$L_1 = \sigma_1 \sum_i L^i_{fea} + \sigma_2 L_{emb} \tag{7}$$

where the hyperparameters $\sigma_1$ and $\sigma_2$ are used to balance the learning of embeddings and features.

### 3.3.2   THE SECOND STAGE: TASK-SPECIFIC DISTILLATION

In stage 2, we take a BERT fine-tuned on a task-specific dataset as the teacher model, and the model completed stage 1 as the student. To accomplish a certain language task, there should be a task-specific head over the basic language model as shown in Figure 2. For example, it is necessary to add an MLP layer over BERT for text classification. Besides, data augmentation is a commonly used and highly effective technique in knowledge distillation(Jiao et al., 2019; Tang et al., 2019; Liu et al., 2022). In the following, we will discuss our approach to data augmentation, as well as the logits loss and cross-entropy loss.

**Data Augmentation**  In the distillation approach, a small dataset may be insufficient for the teacher model to fully express its knowledge(Ba & Caruana, 2013). To tackle this issue, we augment the training set in order to facilitate effective knowledge distillation. We follow Tang et al. (2019) to augment the training set: Firstly, we randomly replace a word with [MASK] token with probability $p_{mask}$. Secondly, we replace a word with another of the same POS tag with probability $p_{pos}$. Thirdly, we randomly sample an $n$-gram from a training example with probability $p_{ng}$, where $n$ is randomly selected from $\{1, 2, ..., 5\}$.

**Logits Loss**  Following Hinton et al. (2015), we take logits, also known as soft labels, into consideration, which lets the student learn the prediction distribution of the teacher. To measure the distance between two distributions, we choose KL-divergence: $L_{logits} = \sum_i^c p_i log \left( \frac{p_i}{q_i} \right)$, where $c$ is the number of categories, $p_i$ and $q_i$ denote the prediction distribution of the teacher model and student model.

**Cross-entropy Loss**  Cross-entropy loss can help the student model learn from the samples in task-specific datasets: $L_{ce} = - \sum_i^c \hat{q}_i log \left( q_i \right)$, where $\hat{q}_i$ represents the one-hot label vector.

Therefore, the total loss $L_2$ of stage 2 contains four terms:

$$L_2 = \lambda_1 \sum_i L^i_{fea} + \lambda_2 L_{emb} + \lambda_3 L_{logits} + \lambda_4 L_{ce} \tag{8}$$

where $\lambda_1$, $\lambda_2$, $\lambda_3$, and $\lambda_4$ are the hype-parameters that control the weight of these loss.

For both stages, we adopt backpropagation through time (BPTT), which is suitable for training spiking neural networks. You can see the detailed derivation in Appendix A if interested.

# 4 EXPERIMENTS

We conduct four sets of experiments. The first is to evaluate the accuracy of SpikeBERT trained with the proposed method on 6 datasets of text classification datasets. The second experiment is to compare the theoretical energy consumption of BERT and that of SpikeBERT. The third experiment is an ablation study about the training process. The last experiment is to figure out how the performance of SpikeBERT is impacted by the number of time steps, model depth, and decay rate.

## 4.1 DATASETS

As mentioned in Section 3.3.1, a large-scale parallel corpus will be used to train student models in Stage 1. For the English corpus, we choose the "20220301.en" subset of Wikipedia and the whole Bookcorpus(Zhu et al., 2015), which are both utilized to pre-train a BERT (Devlin et al., 2019). For the Chinese corpus, we choose Chinese-Wikipedia dump (as of Jan. 4, 2023). Additionally, we follow Lv et al. (2023) to evaluate the SpikeBERT trained with the proposed distillation method on six text classification datasets: MR(Pang & Lee, 2005), SST-2(Socher et al., 2013), SST-5, Subj, ChnSenti, and Waimai. The dataset details are provided in Appendix B.

## 4.2 IMPLEMENTATION DETAILS

Firstly, we set the number of encoder blocks in SpikeBERT to 12. Additionally, we set the threshold of common spiking neurons $U_{thr}$ as 1.0 but set the threshold of neurons in the spiking self-attention block as 0.25 in SpikeBERT. In addition, we set decay rate $\beta = 0.9$ and scaling factor $\tau$ as 0.125. We also set the time step $T$ of spiking inputs as 4 and sentence length to 256 for all datasets. To construct SpikeBERT, we use two Pytorch-based frameworks: SnnTorch (Eshraghian et al., 2021) and SpikingJelly (Fang et al., 2020a). Besides, we utilize bert-base-cased from Huggingface as teacher model for English datasets and Chinese-bert-wwm-base (Cui et al., 2019) for Chinese datasets. In addition, we conduct pre-training distillation on 4 NVIDIA A100-PCIE GPUs and task-specific distillation on 4 NVIDIA GeForce RTX 3090 GPUs. Since surrogate gradients are required during backpropagation, we set $\alpha$ in Equation 3 as 2. In stage 1, we set the batch size as 128 and adopt AdamW (Loshchilov & Hutter, 2017) optimizer with a learning rate of $5e^{-4}$ and a weight decay rate of $5e^{-3}$. The hyperparameters $\sigma_1$ and $\sigma_2$ in Equation 7 are both set to 1.0. In stage 2, we set the batch size as 32 and the learning rate to $5e^{-5}$. For data augmentation, we set $p_{mask} = p_{pos} = 0.1$, $p_{ng} = 0.25$. To balance the weights of the four types of loss in Equation 8, we set $\lambda_1 = 0.1$, $\lambda_2 = 0.1$, $\lambda_3 = 1.0$, and $\lambda_4 = 0.1$.

## 4.3 MAIN RESULTS

We report in Table 1 the accuracy achieved by SpikeBERT trained with "pre-training + task-specific" distillation on 6 datasets, compared to 2 baselines: 1) SNN-TextCNN proposed by Lv et al. (2023); 2) improved Spikformer directly trained with gradient descent algorithm using surrogate gradients. Meanwhile, we report the performance of SpikeBERT on the GLUE benchmark in Appendix C.

Table 1: Classification accuracy achieved by different methods on 6 datasets. A BERT model fine-tuned on the dataset is denoted as "FT BERT". The improved Spikformer directly trained with surrogate gradients on the dataset is denoted as "Directly-trained Spikformer". All reported experimental results are averaged across 10 random seeds.

| Model | English Dataset | | | | Chinese Dataset | | Avg. |
|---|---|---|---|---|---|---|---|
| | MR | SST-2 | Subj | SST-5 | ChnSenti | Waimai | |
| TextCNN (Kim, 2014) | $77.41_{\pm0.22}$ | $83.25_{\pm0.16}$ | $94.00_{\pm0.22}$ | $45.48_{\pm0.16}$ | $86.74_{\pm0.15}$ | $88.49_{\pm0.16}$ | 79.23 |
| FT BERT (Devlin et al., 2019) | $87.63_{\pm0.18}$ | $92.31_{\pm0.17}$ | $95.90_{\pm0.16}$ | $50.41_{\pm0.13}$ | $89.48_{\pm0.16}$ | $90.27_{\pm0.13}$ | **84.33** |
| SNN-TextCNN (Lv et al., 2023) | $75.45_{\pm0.51}$ | $80.91_{\pm0.34}$ | $90.60_{\pm0.32}$ | $41.63_{\pm0.44}$ | $85.02_{\pm0.22}$ | $86.66_{\pm0.17}$ | 76.71 |
| Directly-trained Spikformer | $76.38_{\pm0.43}$ | $81.55_{\pm0.28}$ | $91.80_{\pm0.29}$ | $42.02_{\pm0.45}$ | $85.45_{\pm0.29}$ | $86.93_{\pm0.20}$ | 77.36 |
| SpikeBERT [Ours] | $\mathbf{80.69_{\pm0.44}}$ | $\mathbf{85.39_{\pm0.36}}$ | $\mathbf{93.00_{\pm0.33}}$ | $\mathbf{46.11_{\pm0.40}}$ | $\mathbf{86.36_{\pm0.28}}$ | $\mathbf{89.66_{\pm0.21}}$ | **80.20** |

Table 1 demonstrates that the SpikeBERT trained with two-stage distillation achieves state-out-of-art performance across 6 text classification datasets. Compared to SNN-TextCNN, SpikeBERT achieved up to 5.42% improvement in accuracy (3.49% increase on average) for all text classification benchmarks. Furthermore, SpikeBERT outperforms TextCNN, which is considered a representative artificial neural network, and even achieves comparable results to the fine-tuned BERT by a small

drop of 4.13% on average in accuracy for text classification task. What's more, Table 1 demonstrates that SpikeBERT can also be applied well in Chinese datasets (ChnSenti and Waimai). Fang et al. (2020b) propose that, in image classification task, surrogate gradients of SNNs may lead to gradient vanishing or exploding and it is even getting worse with the increase of model depth. We found this phenomenon in language tasks as well. Table 1 reveals that the accuracy of directly-trained Spikformer is noticeably lower than SpikeBERT on some benchmarks, such as MR, SST-5, and ChnSenti. This is likely because the directly-trained Spikformer models have not yet fully converged due to gradient vanishing or exploding.

## 4.4 ENERGY CONSUMPTION

An essential advantage of SNNs is the low consumption of energy during inference. Assuming that we run SpikeBERT on a 45nm neuromorphic hardware (Horowitz, 2014), we are able to calculate the theoretical energy consumption based on Appendix D. We compare the theoretical energy consumption per sample of fine-tuned BERT and SpikeBERT on 6 test datasets and report the results in Table 2.

Table 2: Energy consumption per sample of fine-tuned BERT and SpikeBERT during inference on 6 text classification benchmarks. "FLOPs" denotes the floating point operations of fine-tuned BERT. "SOPs" denotes the synaptic operations of SpikeBERT.

| Dataset | Model | FLOPs / SOPs(G) | Energy (mJ) | Energy Reduction | Accuracy (%) |
|---|---|---|---|---|---|
| ChnSenti | FT BERT | 22.46 | 103.38 | **70.49**% ↓ | 89.48 |
| | SpikeBERT | 28.47 | 30.51 | | 86.36 |
| Waimai | FT BERT | 22.46 | 103.38 | **71.08**% ↓ | 90.27 |
| | SpikeBERT | 27.81 | 29.90 | | 89.66 |
| MR | FT BERT | 22.23 | 102.24 | **72.58**% ↓ | 87.63 |
| | SpikeBERT | 26.94 | 28.03 | | 80.69 |
| SST-2 | FT BERT | 22.23 | 102.24 | **72.09**% ↓ | 92.31 |
| | SpikeBERT | 27.46 | 28.54 | | 85.39 |
| Subj | FT BERT | 22.23 | 102.24 | **73.63**% ↓ | 95.90 |
| | SpikeBERT | 25.92 | 26.96 | | 93.00 |
| SST-5 | FT BERT | 22.23 | 102.24 | **73.27**% ↓ | 50.41 |
| | SpikeBERT | 26.01 | 27.33 | | 46.11 |

As shown in Table 2, the energy consumption of SpikeBERT is significantly lower than that of fine-tuned BERT, which is an important advantage of SNNs over ANNs in terms of energy efficiency. SpikeBERT demands only 27.82% of the energy that fine-tuned BERT needs to achieve comparable performance on average. This indicates that SpikeBERT is a promising candidate for energy-efficient text classification in resource-constrained scenarios. It is worth noting that energy efficiency achieved by spiking neural networks (SNNs) is distinct from model compressing methods such as knowledge distillation or model pruning. They represent different technological pathways. Spiking neural networks do not alter the model parameters but instead introduce temporal signals to enable the model to operate in a more biologically plausible manner on neuromorphic hardware. The energy reduction of spiking neural networks is still an estimate, and future advancements in hardware are expected to decrease energy consumption further while potentially accelerating inference speeds. We show the comparison of energy reduction between SpikeBERT and other BERT variants, such as TinyBERT (Jiao et al., 2019) and DistilBERT (Sanh et al., 2019) in Appendix E.

## 4.5 ABLATION STUDY AND IMPACT OF HYPER-PARAMETERS

In this section, we conduct ablation studies to investigate the contributions of: a) different stages of the proposed knowledge distillation method, and b) different types of loss in Equation 8.

As we can see in Table 4.5, SpikeBERTs without either stage 1 or stage 2 experience about 3.20% performance drop on average. Therefore, we conclude that the two distillation stages are both essential for training SpikeBERT. Furthermore, we observed that the average performance dropped from 76.30 to 73.27 when excluding the logits loss, demonstrating that the logits loss $L_{logits}$ has the greatest impact on task-specific distillation. Meanwhile, data augmentation (DA) plays an important role in Stage 2, contributing to an increase in average performance from 75.54 to 76.30.

Table 3: Ablation studies of the two-stage distillation method. Row 3 and 4 show ablation experiment results on the two steps of our proposed method. Row 5 to 9 are ablation experiment results on different parts of Equation 8. "DA" stands for data augmentation.

| | Models | MR | SST-2 | Subj | SST-5 | Avg. | Drop |
|---|---|---|---|---|---|---|---|
| | SpikeBERT | 80.69 | 85.39 | 93.00 | 46.11 | 76.30 | $--$ |
| | w/o Stage 1 | 76.04 | 82.26 | 91.80 | 42.16 | 73.07 | $-3.23$ |
| | w/o Stage 2 | 75.91 | 82.26 | 91.90 | 42.58 | 73.14 | $-3.16$ |
| | w/o DA | 80.22 | 84.90 | 92.20 | 44.84 | 75.54 | $-0.76$ |
| | w/o $L_{fea}$ | 78.35 | 83.48 | 92.20 | 43.57 | 74.40 | $-1.90$ |
| Stage 2 | w/o $L_{emb}$ | 79.67 | 83.10 | 92.00 | 43.48 | 74.56 | $-1.74$ |
| | w/o $L_{logits}$ | 76.19 | 82.64 | 91.90 | 42.35 | 73.27 | $-3.03$ |
| | w/o $L_{ce}$ | 80.43 | 85.23 | 93.00 | 45.86 | 76.13 | $-0.17$ |

We investigate how the performance of SpikeBERT is affected by the two important hyperparameters: time steps $T$ and model depth. To this end, we conduct three experiments: (a) varying the number of the time steps of spike inputs when training SpikeBERT; (b) training a variant of SpikeBERT with different encoder block depths, specifically $6, 12, 18$, using our proposed two-stage method; (c) varying the decay rate $\beta$ when training SpikeBERT;

Figure 3 (a) shows how the accuracy of SpikeBERT varies with the increase of time steps. We find that, with the increase of time steps, the accuracy increases first, then remains unchanged, and reaches its maximum roughly at $T = 4$. Theoretically, the performance of SpikeBERT should be higher with bigger time steps. However, the performance of models with $8$ and $12$ time steps is even worse than that with $4$ time steps on ChnSenti and Waimai datasets. A plausible explanation is that using excessively large time steps may introduce too much noise in the spike trains.

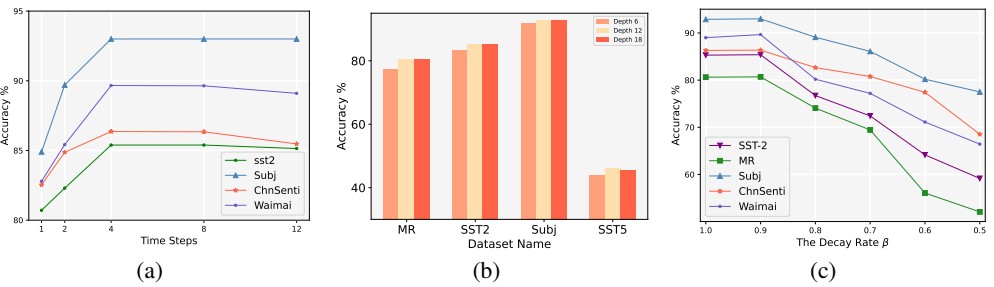

(a)  (b)  (c)

Figure 3: (a) Accuracy versus the number of time steps. (b) Accuracy versus the depth of networks. (c) Accuracy versus the decay rate $\beta$.

In addition, as we can see from Figure 3 (b), the accuracy of SpikeBERT is generally insensitive to the model depths and even gets lower in some datasets. We think that's because more spike Transformer blocks bring more spiking neurons, which introduce more surrogate gradients when error backpropagation through time. It seems that deeper spiking neural networks cannot make further progress in performance. Many previous SNNs works (Zheng et al., 2020; Fang et al., 2020b; Kim et al., 2022a) have proved this deduction. From Figure 3 (c), we find that decay rate $\beta$ cannot be taken too small, or too much information will be lost.

## 5  CONCLUSION

In this study, we extended and improved Spikformer to process language tasks and proposed a new promising training paradigm for training SpikeBERT inspired by the notion of knowledge distillation. We presented a two-stage, "pre-training + task-specific" knowledge distillation method by transferring the knowledge from BERTs to SpikeBERT for text classification tasks. We empirically show that our SpikeBERT outperforms the state-of-the-art SNNs and can even achieve comparable results to BERTs with much less energy consumption across multiple datasets for both English and Chinese, leading to future energy-efficient implementations of BERTs or large language models. The limitations of our work are discussed in Appendix F.

## REPRODUCIBILITY STATEMENT

The authors have made great efforts to ensure the reproducibility of the empirical results reported in this paper. The experiment settings, evaluation metrics, and datasets were described in detail in Subsections 4.1, 4.2, and Appendix B. Additionally, we had submitted the source code of the proposed training algorithm with our paper, and plan to release the source code on GitHub upon acceptance.

## ETHICS STATEMENT

We think this paper will not raise questions regarding the Code of Ethics.

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

APPENDIX

## A    BACKPROPAGATION THROUGH TIME IN SPIKING NEURAL NETWORKS

The content of this section is mostly referred to Lv et al. (2023).

Given a loss function $L$ like Equation 7 and 8, the losses at every time step can be summed together to give the following global gradient:

$$\frac{\partial L}{\partial W} = \sum_t \frac{\partial L_t}{\partial W} = \sum_i \sum_{j \leq i} \frac{\partial L_i}{\partial W_j} \frac{\partial W_j}{\partial W} \tag{9}$$

where $i$ and $j$ denote different time steps, and $L_t$ is the loss calculated at time step $t$. No matter which time step is, the weights of an SNN are shared across all steps. Therefore, we have $W_0 = W_1 = \cdots = W$, which also indicates that $\frac{\partial W_j}{\partial W} = 1$. Thus, Equation (9) can be written as follows:

$$\frac{\partial L}{\partial W} = \sum_i \sum_{j \leq i} \frac{\partial L_i}{\partial W_j} \tag{10}$$

Based on the chain rule of derivatives, we obtain:

$$\begin{aligned}
\frac{\partial L}{\partial W} &= \sum_i \sum_{j \leq i} \frac{\partial L_i}{\partial S_i} \frac{\partial S_i}{\partial U_i} \frac{\partial U_i}{\partial W_j} \\
&= \sum_i \frac{\partial L_i}{\partial S_i} \frac{\partial S_i}{\partial U_i} \sum_{j \leq i} \frac{\partial U_i}{\partial W_j}
\end{aligned} \tag{11}$$

where $\frac{\partial L_i}{\partial S_i}$ is the derivative of the cross-entropy loss at the time step $i$ with respect to $S_i$, and $\frac{\partial S_i}{\partial U_i}$ can be easily derived using surrogate gradients like Equation 3. As to the last term of $\sum_{j \leq i} \frac{\partial U_i}{\partial W_j}$, we can split it into two parts:

$$\sum_{j \leq i} \frac{\partial U_i}{\partial W_j} = \frac{\partial U_i}{\partial W_i} + \sum_{j \leq i-1} \frac{\partial U_i}{\partial W_j} \tag{12}$$

From Equation (2), we know that $\frac{\partial U_i}{\partial W_i} = X_i$. Therefore, Equation (9) can be simplified as follows:

$$\frac{\partial L}{\partial W} = \sum_i \underbrace{\frac{\partial L_i}{\partial S_i} \frac{\partial S_i}{\partial U_i}}_{\text{constant}} \left( \underbrace{\frac{\partial U_i}{\partial W_j}}_{\text{constant}} + \sum_{j \leq i-1} \frac{\partial U_i}{\partial W_j} \right) \tag{13}$$

By the chain rule of derivatives over time, $\frac{\partial U_i}{\partial W_j}$ can be factorized into two parts:

$$\frac{\partial U_i}{\partial W_j} = \frac{\partial U_i}{\partial U_{i-1}} \frac{\partial U_{i-1}}{\partial W_j} \tag{14}$$

It is easy to see that $\frac{\partial U_i}{\partial U_{i-1}}$ is equal to $\beta$ from Equation (2), and Equation (9) can be written as:

$$\frac{\partial L}{\partial W} = \sum_i \underbrace{\frac{\partial L_i}{\partial S_i} \frac{\partial S_i}{\partial U_i}}_{\text{constant}} \left( \underbrace{\frac{\partial U_i}{\partial W_j}}_{\text{constant}} + \sum_{j \leq i-1} \underbrace{\frac{\partial U_i}{\partial U_{i-1}}}_{\text{constant}} \frac{\partial U_{i-1}}{\partial W_j} \right) \tag{15}$$

We can treat $\frac{\partial U_{i-1}}{\partial W_j}$ recurrently as Equation (12). Finally, we can update the weights $W$ by the rule of $W = W - \eta \frac{\partial L}{\partial W}$, where $\eta$ is a learning rate.

## B    DATASETS

The benchmark we used in Table 1 includes the following datasets:

- **MR**: MR stands for Movie Review and it consists of movie-review documents labeled with respect to their overall sentiment polarity (positive or negative) or subjective rating (Pang & Lee, 2005).
- **SST-5**: SST-5 contains $11,855$ sentences extracted from movie reviews for sentiment classification (Socher et al., 2013). There are $5$ categories (very negative, negative, neutral, positive, and very positive).
- **SST-2**: The binary version of SST-5. There are just $2$ classes (positive and negative).
- **Subj**: The task of this dataset is to classify a sentence as being subjective or objective[1].
- **ChnSenti**: ChnSenti comprises about $7,000$ Chinese hotel reviews annotated with positive or negative labels[2].
- **Waimai**: There are about $12,000$ Chinese user reviews collected by a food delivery platform for binary sentiment classification (positive and negative)[3] in this dataset.

## C  PERFORMANCE ON GLUE

General Language Understanding Evaluation (GLUE) benchmark is a collection of diverse natural language understanding tasks. We report the performance of SpikeBERT on GLUE benchmark on Table 4

Table 4: Classification accuracy achieved by different models on the GLUE benchmark. A BERT model fine-tuned on the dataset is denoted as "FT BERT". "SNN-TextCNN" is a SNN baseline proposed by Lv et al. (2023). $*$ indicates that the model fails to converge. All reported experimental results are averaged across 10 random seeds.

| Task | SST-2 | MRPC | RTE | QNLI | MNLI-(m/mm) | QQP | CoLA | STS-B |
|------|-------|------|-----|------|-------------|-----|------|-------|
| Metric | Acc | F1 | Acc | Acc | acc | F1 | Matthew's corr | Spearman's corr |
| FT BERT | 92.31 | 89.80 | 69.31 | 90.70 | 83.82/83.41 | 90.51 | 60.00 | 89.41 |
| SNN-TextCNN | 80.91 | 80.62 | 47.29* | 56.23* | 64.91/63.69 | 0.00* | $-5.28$* | 0.00* |
| SpikeBERT | 85.39 | 81.98 | 57.47 | 66.37 | 71.42/70.95 | 68.17 | 16.86* | 18.73* |

Although SpikeBERT significantly outperforms the SNN baseline on all tasks, we find that the performance of SpikeBERT on the Natural Language Inference (NLI) task (QQP, QNLI, RTE) is not satisfactory compared to fine-tuned BERT. The possible reason is that we mainly focus on the semantic representation of a single sentence in the pre-training distillation stage. Meanwhile, we have to admit that SpikeBERT is not sensitive to the change of certain words or synonyms, for it fails to converge on CoLA and STS-B datasets. We think that's because spike trains are much worse than floating-point data in representing fine-grained words. In the future, we intend to explore the incorporation of novel pre-training loss functions to enhance the model's ability to model sentence entailment effectively.

## D  THEORETICAL ENERGY CONSUMPTION CALCULATION

According to Yao et al. (2022), for spiking neural networks (SNNs), the theoretical energy consumption of layer $l$ can be calculated as:

$$Energy(l) = E_{AC} \times SOPs(l) \tag{16}$$

where SOPs is the number of spike-based accumulate (AC) operations. For traditional artificial neural networks (ANNs), the theoretical energy consumption required by the layer $b$ can be estimated by

$$Energy(b) = E_{MAC} \times FLOPs(b) \tag{17}$$

---

[1] https://www.cs.cornell.edu/people/pabo/movie-review-data/

[2] https://raw.githubusercontent.com/SophonPlus/ChineseNlpCorpus/master/datasets/ChnSentiCorp_htl_all/ChnSentiCorp_htl_all.csv

[3] https://raw.githubusercontent.com/SophonPlus/ChineseNlpCorpus/master/datasets/waimai_10k/waimai_10k.csv

where FLOPs is the floating point operations of $b$, which is the number of multiply-and-accumulate (MAC) operations. We assume that the MAC and AC operations are implemented on the 45nm hardware (Horowitz, 2014), where $E_{MAC} = 4.6pJ$ and $E_{AC} = 0.9pJ$. Note that $1J = 10^3$ mJ $= 10^{12}$ pJ. The number of synaptic operations at the layer $l$ of an SNN is estimated as

$$SOPs(l) = T \times \gamma \times FLOPs(l) \tag{18}$$

where $T$ is the number of times step required in the simulation, $\gamma$ is the firing rate of input spike train of the layer $l$.

Therefore, we estimate the theoretical energy consumption of SpikeBERT as follows:

$$E_{SpikeBERT} = E_{MAC} \times EMB_{emb}^1 + E_{AC} \times \left( \sum_{m=1}^{M} \text{SOP}_{\text{SNN FC}}^m + \sum_{n=1}^{N} \text{SOP}_{\text{SSA}} \right) \tag{19}$$

where $EMB_{emb}^1$ is the embedding layer of SpikeBERT. Then the SOPs of $m$ SNN Fully Connected Layer (FC) and $l$ SSA are added together and multiplied by $E_{AC}$.

## E  ENERGY REDUCTION COMPARED TO OTHER BERT VARIANTS

We compare the energy reduction between SpikeBERT, Tiny BERT(Jiao et al., 2019), and Distil-BERT(Sanh et al., 2019) in Table 5.

Table 5: Energy consumption per sample of fine-tuned BERT, SpikeBERT, TinyBERT and DistilBERT during inference on 3 text classification benchmarks. "FLOPs" denotes the floating point operations of ANNs. "SOPs" denotes the synaptic operations of SpikeBERT. "Energy" denotes the average theoretical energy required for each test example prediction.

| Dataset | Model | Parameters(M) | FLOPs/SOPs(G) | Energy(mJ) | Energy Reduction | Accuracy(%) |
|---------|-------|---------------|---------------|------------|------------------|-------------|
| Waimai | FT BERT | 109.0 | 22.46 | 103.38 | - | 90.27 |
| | SpikeBERT | 109.0 | 27.81 | 29.90 | 71.08%↓ | 89.66 |
| | TinyBERT | 67.0 | 11.30 | 52.01 | 49.69%↓ | 89.72 |
| | DistilBERT | 52.2 | 7.60 | 34.98 | 66.16%↓ | 89.40 |
| ChnSenti | FT BERT | 109.0 | 22.46 | 103.38 | - | 89.48 |
| | SpikeBERT | 109.0 | 28.47 | 30.51 | 70.49%↓ | 86.36 |
| | TinyBERT | 67.0 | 11.30 | 52.01 | 49.69%↓ | 88.70 |
| | DistilBERT | 52.2 | 7.60 | 34.98 | 66.16%↓ | 87.41 |
| SST-2 | FT BERT | 109.0 | 22.23 | 102.24 | - | 92.31 |
| | SpikeBERT | 109.0 | 27.46 | 28.54 | 72.09%↓ | 85.39 |
| | TinyBERT | 67.0 | 11.30 | 52.01 | 49.13%↓ | 91.60 |
| | DistilBERT | 52.2 | 7.60 | 34.98 | 65.78%↓ | 90.40 |

We want to state again that spiking neural networks and model compressing are two different technological pathways to achieve energy efficiency. Future advancements in neuromorphic hardware are expected to decrease energy consumption further.

## F  DISCUSSION OF LIMITATIONS

In the image classification task, spiking neural networks have demonstrated comparable performance to ViT on CIFAR-10-DVS and DVS-128-Gesture datasets, which are neuromorphic event-based image datasets created using dynamic vision sensors. We think that the performance gap between SNNs and ANNs in language tasks is mainly due to the lack of neuromorphic language datasets. It is unfair to evaluate SNNs on the datasets that were created to train and evaluate ANNs because these datasets are mostly processed by continuous values. However, it is quite hard to convert language to neuromorphic information without information loss. We hope there will be a new technology to transfer sentences to neuromorphic spikes.

In addition, GPU memory poses a limitation in our experiments. Spiking neural networks have an additional dimension, denoted as $T$ (time step), compared to artificial neural networks. Increasing the number of time steps allows for capturing more information but results in an increased demand for GPU memory by a factor of $T$. During our experiments, we observe that maintaining the same number of time steps during training requires reducing the sentence length of input sentences, which significantly constrains the performance of our models. We remain optimistic that future advancements will provide GPUs with sufficient memory to support the functionality of SNNs.

## G    VISUALIZATIONS OF ATTENTION MAP IN SPIKEBERT

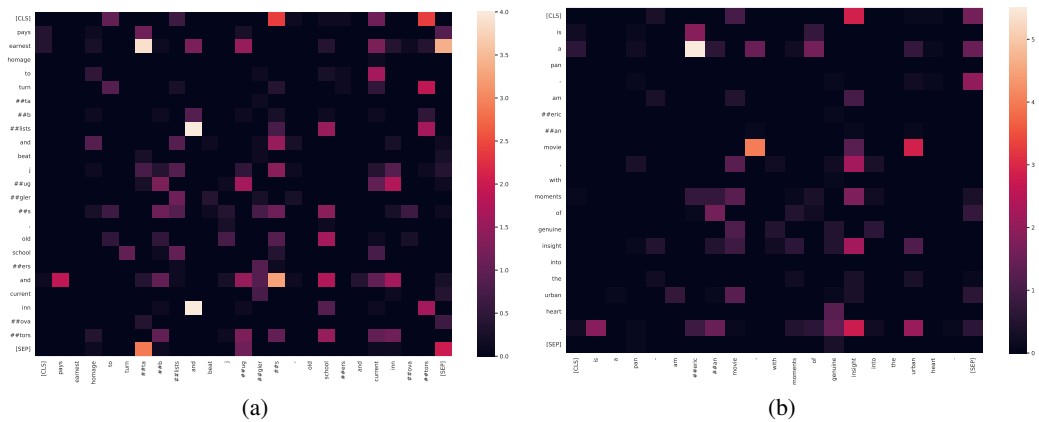

(a)                                                              (b)

Figure 4:  Attention map examples of SSA. (a) Content of the sample is: "pays earnest homage to turntablists and beat jugglers, old schoolers and current innovators". It is positive. (b) Content of the sample is: "is a pan-american movie, with moments of genuine insight into the urban heart .". It is positive.

We show the attention map examples of the last encoder block in SpikeBERT at the last time step in Figure 4. In Figure 4 (a), the positive token "earnest" receives the most attention, and in Figure 4 (b), most tokens focus on the "insight" token, which can be seen as positive. Therefore, we can conclude that the spiking self-attention module can successfully capture semantics at the word level associated with classification semantics, and is shown to be effective and event-driven.

## H    MOTIVATIONS OF OUR STUDY

### H.1    WHAT IS SPIKING NEURAL NETWORK? WHY IS IT IMPORTANT?

Spiking neural network (SNN) is a brain-inspired neural network proposed by Maas (1997), which has been seen as the third generation of neural network models and an important application of neuroscience SNNs use discrete spike trains (0 and 1 only) instead of floating-point values to compute and transmit information, which are quite suitable for implementation on neuromorphic hardware. Therefore, compared to traditional artificial neural networks that run on GPUs, the SNNs offer an energy-efficient computing paradigm to deal with large volumes of data using spike trains for information representation when inference. To some degree, we can regard SNN as a simulation of neuromorphic hardware used to handle a downstream deep-learning task. Nowadays, neuromorphic hardware mainly refers to brain-like chips, such as 14nm Loihi2 (Intel), 28nm TrueNorth (IBM), etc. As for the training of SNNs, there are no mature on-chip training solutions currently so the training has to be done on GPUs. However, once SNNs are well-trained on GPUs, they can be deployed on neuromorphic hardware for energy-efficient computing (0-1 computing only).

### H.2    OUR MOTIVATIONS

There are few works that have demonstrated the effectiveness of spiking neural networks in natural language processing tasks. The current state-of-the-art model is SNN-TextCNN(Lv et al., 2023),

which is based on a simple backbone TextCNN. However, LLMs like ChatGPT perform very well in many language tasks nowadays, but one of their potential problems is the huge energy consumption (even when inference with GPUs). We want to implement spiking versions of LLMs running on low-energy neuromorphic hardware so that models are able to do GPUs-free inference and the energy consumption when inference can be significantly reduced. Our proposed SpikeBERT can be seen as the first step. We hope SpikeBERT can lead to future energy-efficient implementations of large language models on brain-inspired neuromorphic hardware. We would like to clarify that our approach involves the challenging task of distilling knowledge from BERT into spiking neural networks (SNNs), where the fundamental distinction lies in the use of discrete spikes for computation and information transmission in the student model (SNN), as opposed to the continuous values in the teacher model (BERT). To address this disparity, we introduce a novel two-stage "pre-training + task-specific" knowledge distillation (KD) method. This method incorporates proper normalization across both timesteps and training instances within a batch, enabling a meaningful comparison and alignment of feature representations between the teacher and student models.

