# OpenReview forum: "SpikeBERT: A Language Spikformer Learned from BERT with Knowledge Distillation"
_ICLR.cc/2024/Conference — ICLR 2024 Conference Withdrawn Submission_

### Official Review · Reviewer_sLn7 · 2023-10-25

**Soundness:** 2 fair
**Presentation:** 2 fair
**Contribution:** 1 poor
**Rating:** 3
**Confidence:** 4

**Summary:**

The authors propose a spiking neural network variant of BERT called SpikeBERT, thereby employing knowledge distillation using BERT as the teacher model. The main advantage of SpikeBERT compared to vanilla BERT seems to be that it is consuming less “energy” (measured in mJ).

**Strengths:**

- **Table 2**: It’s interesting and promising to see that SpikeBERT has much lower energy consumption than FT BERT.


- **Section 4.5**: I appreciate the comprehensive ablation study that was performed on the hyperparameters. It’s good scientific practice to scrutinize the impact/effect of different hyperparameters.

**Weaknesses:**

- **Contributions**: I think contribution 2 is misleading. While the authors show that their model performs better than existing SNN methods, their method performs on par with a simple TextCNN (from **ten** years ago) which I think is computationally much less expensive than their whole pre-training and fine-tuning pipeline because TextCNN is a much smaller architecture whose outputs are non-contextual/static word representations that can easily be pre-computed.

- **Section 3**: The entire “pre-training + knowledge distillation + fine-tuning” pipeline appears to require a vanilla BERT model that has previously been pretrained on a large language corpus as is standard. If you rely on BERT, why would I not “just” fine-tune or probe BERT instead of applying your pipeline? What is the advantage here?


- **Notation/Math**: The math and notation in Section 3 are a bit sloppy and not very precise.


- **Table 1**: The results in Table 1 are misleading. The authors bold-faced their model’s performances. However, “FT BERT” (which is clearly not SOTA anymore on these tasks) achieves much stronger performance than their model across all reported datasets. Moreover, TextCNN --- which was one of the **first** CNN models for text sequences and whose representations are non-contextual/static word representations --- shows better performance on two datasets (Subj and ChnSenti) and only marginally worse performance on the other four datasets. I’d be curious to see the standard deviations here. Because, if they overlap, then the performances between TextCNN and SpikeBERT are not statistically significantly different. Please report the standard deviations in brackets next to the averages and unbold your numbers or at least explain in the caption what bold-face means here. It’s not good practice to mislead the reader by simply bold-facing your numbers without further explanation.


- **Conclusion**: The conclusion is pretty short for a scientific conference paper. There is no discussion of results or impact. Moreover, I think the claim “*[...] can even achieve comparable results to BERTs with much less energy consumption across multiple datasets for both English and Chinese, leading to future energy-efficient implementations of BERTs or large language models.*” is misleading. I think SpikeBERT achieves comparable results to TextCNN but not to FT BERT. Also, BERT is not SOTA anymore since 2021. How would SpikeBERT compare against more recent variants of Transformer-based foundation models such as RoBERTa, Albert, or T5? (see the [Glue](https://gluebenchmark.com/), [SuperGlue](https://super.gluebenchmark.com/) and [SQuAD](https://rajpurkar.github.io/SQuAD-explorer/) leaderboards for an up-to-date list of models in NLP). I am not convinced that the approach reported in this paper is “*leading to future energy-efficient implementations of BERTs or large language models*”. There are numerous other approaches that have demonstrated this via distillation techniques (e.g., [DistilBERT](https://huggingface.co/docs/transformers/model_doc/distilbert)).


- No limitations are discussed as part of the conclusion. Unfortunately, there is no discussion section.


- An entire body of work that has employed distillation techniques over the past 4 years in NLP is not discussed here.

**Questions:**

- **Section 3**: The entire “pre-training + knowledge distillation + fine-tuning” pipeline appears to require a vanilla BERT model that has previously been pretrained on a large language corpus as is standard. If you rely on BERT, why would I not “just” fine-tune or probe BERT instead of applying your pipeline? What is the advantage here? Linear probing is much less expensive than fine-tuning (it only requires a linear classifier) and often equally performant (depending on the task).


- **Section 4**: Could you elaborate why I would use SpikeBERT over TextCNN although the methods perform equally well on all the reported datasets (see my comment on Table 1 above)? TextCNN is a computationally much less expensive method and has the "advantage" of static word representations. So, I could just compute the representations for each word used in the datasets a priori and then run inference as many times as I want without the need to run the sentences through the model. That being said, I don’t think that anyone in the community would still use a TextCNN from 2013 that produces non-contextual word representations for NLP tasks.


- Why didn’t you compare SpikeBERT against [DistilBERT](https://huggingface.co/docs/transformers/model_doc/distilbert)? DistilBERT is a distilled version of BERT that is much faster and cheaper and has comparable performance to BERT (probably better than SpikeBERT on the benchmarks that you looked at). AFAIK, DistilBERT exists since 2020. So, there must be an even better and more recent version of DistilBERT such as DistilRoBERTa or DistilALBERT. But please take a look yourself.


- **Table 2**: Why is the number of FLOPs consistently larger for SpikeBERT than for FT BERT although SpikeBERT’s energy consumption is much lower? How do you explain that? I’d like to see FLOPs and energy consumption of TextCNN reported in this table and not just SpikeBERT vs. FT BERT. Could you please report those? In addition to the FLOPs and energy consumption of TextCNN it would like to see these numbers for [DistilBERT](https://huggingface.co/docs/transformers/model_doc/distilbert) which has been shown to be much more computationally efficient/energy efficient than BERT while preserving most of its performance (around 95%) via knowledge distillation but their pipeline seems to be easier than your pipeline and does not necessitate "embedding alignment". Again, there probably exist even better distilled versions of BERT or RoBERTa or GPT-2/GPT-3 by now.


- **Table 3**: Did you employ the same data augmentation strategies to all methods that you compared SpikeBERT against? If data augmentation plays a crucial role in the performance of the model (which it does according to your ablations), then it seems not fair to compare SpikeBERT + DA against other methods without DA.

---

> ### Author Response · Authors · 2023-11-13
>
> Thank you for your valuable comments!
>
> **Q1:** **If you rely on BERT, why would I not “just” fine-tune or probe BERT instead of applying your pipeline? What is the advantage here? Linear probing is much less expensive than fine-tuning (it only requires a linear classifier) and often equally performant (depending on the task).**
>
> A1: It is our fault that we do not provide enough background on spiking neural networks.
>
> 1) Different from classical artificial neural networks, spiking neural networks are **brain-inspired** networks, which do not transmit information in form of continuous values, but rather the time when a membrane potential reaches a specific threshold. Once the membrane potential reaches the threshold, the neuron fires and generates a pulse signal that travels to the downstream neurons which increase or decrease their potentials in proportion to the connection strengths in response to this signal. SNNs incorporate the concept of time into their computing model in addition to neuronal and synaptic states. They are considered to be more **biologically plausible** neuronal models than classical ANNs. Besides, SNNs are suitable for implementation on low-power brain-inspired neuromorphic hardware, and offer a promising computing paradigm to deal with large volumes of data using spike trains for information representation in a more **energy-efficient** manner.
>
> 2) We treat SNNs as a promising avenue to implement deep neural networks in a more biologically plausible and energy-efficient (when inference) way. Once SNNs are well software-trained, they can be deployed on neuromorphic hardware (such as 45nm neuromorphic hardware[6]) for energy-efficient computing (binary-value computing). This computing pattern brings a significant energy reduction when inference (Please see Table 2).
>
> 3) Our claim is to use **discrete spike trains** instead of continuous decimal values to compute and transmit information in deep neural networks for language tasks. We hope our work can take a step forward in language processing with a more brain-inspired and energy-efficient method. Our proposed SpikeBERT is the first Transformer-liked SNN for language tasks, and now we are focusing on how to implement spiking version of GPT-liked language model, which may lead to future energy-efficient implementations of large language models **running on neuromorphic** **hardware**.
>
> Therefore, “just” fine-tuning or probing BERTs, which run on GPUs, can not achieve our targets in this scenario. We hope our explanations can address your concerns. Thank you!
>
> **Q2: Contribution 2 is misleading. I think SpikeBERT achieves comparable results to TextCNN but not to FT BERT.**
>
> A2: We apologize that our statement has mislead you. We follow previous work[1][2] to take spiking neural networks (SNNs) as our baselines, rather than traditional artificial neural networks (ANNs). We think it is not that reasonable to take ANNs as baselines, and the reasons are following: 1) Due to the non-differentiability of spikes, training SNNs is a great challenge; 2) ANNs take a large number of floating-point operations, while SNNs only do spike binary-value (only 0 and 1) operations. Therefore, we take SNN-TextCNN and Directly-trained SpikeBERT as our baselines.
>
> This point is widely recognized in the field of SNNs. For example, the performance gap between Spikformer[1] and its ANN counterpart model Vision-Transformer[3] reached 6%, which is much more than the gap between fine-tuned BERT and SpikeBERT. But Spikformer outperformed other SNN models. They also said that their model achieved state-of-the-art performance in SNNs and also comparable to Vision-Transformer. Other SNN papers [2][4][5] also take SNNs as the baselines, rather than ANNs.
>
> Moreover, there are very few works that have demonstrated SNNs’ effectiveness in natural language processing (NLP) tasks. Our proposed SpikeBERT is the first deep Transformer-liked SNN for language tasks, whose scale is the same as the scale of BERT.
>
> **Q3:** **Table 1: The results in Table 1 are misleading.** **Please report the standard deviations in brackets next to the averages and unbold your numbers or at least explain in the caption what bold-face means here.**
>
> A3: It is a good suggestion. As mentioned in Q2, we do not take ANNs as our baselines because it is not fair. However, we will follow your suggestion to report the standard deviations in Table 1 and bold the number of Fine-tuned BERT. Thank you for pointing it out, and we will make it clear in the revised version.

---

> ### Author Response · Authors · 2023-11-13
>
> **Q4:** **BERT is not SOTA anymore since 2021. How would SpikeBERT compare against more recent variants of Transformer-based foundation models such as RoBERTa, Albert, or T5? (see the Glue, SuperGlue and SQuAD leaderboards for an up-to-date list of models in NLP).**
>
> A4: Our approach is applicable to any model similar to BERT or RoBERTa. Existing literature[7] indicates that BERT and RoBERTa exhibit minimal substantive differences in downstream tasks, and both models share a similar network architecture. The divergence between these models is primarily observed in the aspect of masking strategy, input tokenization and training strategy[8], yet these differences do not impact the conclusions drawn in this paper. What’s more, as BERT serves as a representative example of large pre-trained models, we chose BERT as the teacher model.
>
> Besides, we report the performance of SpikeBERT on GLUE benchmark in **Appendix C**. Although SpikeBERT significantly outperforms the SNN baseline (SNN-TextCNN) on all tasks, we find that the performance of SpikeBERT on the Natural Language Inference (NLI) task (QQP, QNLI, RTE) is not satisfactory compared to fine-tuned BERT. The possible reason is that we mainly focus on the semantic representation of a single sentence in the pre-training distillation stage. Meanwhile, we have to admit that SpikeBERT is not sensitive to the change of certain words or synonyms, for it fails to converge on CoLA and STS-B datasets. We think that’s because spike trains are much worse than floating-point data in representing fine-grained words. In the future, we intend to explore the incorporation of novel pre-training loss functions to enhance the model's ability to model sentence entailment effectively.
>
> **Q5:** **No limitations are discussed as part of the conclusion. There is no discussion section.**
>
> A5: Due to page limitations, we have included “Discussion of Limitations” in the **Appendix F** of our original manuscript. In this section, we primarily discuss the following limitations: Firstly, there are many neuromorphic event-based image datasets, such as CIFAR-10-DVS and DVS-128-Gesture, which perfectly align with the characteristics of SNN networks. However, such datasets are lacking in the natural language processing tasks. Secondly, the data used for SNN training introduces an additional temporal dimension (T dimension) compared to traditional data. Limited to the GPU memories, we had to reduce the sentence length of input sentences, which significantly constrains the performance of our models.
>
> **Q6:** **Could you elaborate why I would use SpikeBERT over TextCNN although the methods perform equally well on all the reported datasets (see my comment on Table 1 above)?**
>
> A6: The same as A1.
>
> **Q7:** **An entire body of work that has employed distillation techniques over the past 4 years in NLP is not discussed here.**
>
> A7: Thank you for pointing it out. In the Related Work section, we discuss the history of knowledge distillation in the era of deep learning, including three kinds of knowledge distillation methods, the combination of SNN and knowledge distillation, etc. However, our proposed distillation method is mainly designed to transfer knowledge from ANNs to SNNs. We can add some studies on knowledge distillation in the NLP field over the last 4 years in the revised version.
>
> Also, while it may seem like a natural idea to use knowledge distillation after direct training fails, applying it to spiking neural networks is still poses significant challenges. Firstly, how to align the **spiking signals** of the student model with the **floating-point signals** of the teacher model? We addressed this issue by introducing an external “MLP+LayerNorm” layer to convert the signals from spikes to floating points. Secondly, training spiking neural networks typically requires specific training techniques to stabilize and accelerate the convergence process, so the traditional knowledge distillation methods may not adapt well to these techniques, resulting in training difficulties or suboptimal performance. We addressed this issue by employing many training tricks, some of which were not explicitly mentioned in the paper. These tricks included dynamically adjusting the alignment signal weight ratios based on loss ratios, selectively ignoring representations from certain layers, and using longer warm-up periods. In practice, achieving a convergent spiking neural network language model is challenging because traditional knowledge distillation methods and SNNs training methods are ineffective in these scenarios.

---

> ### Author Response · Authors · 2023-11-13
>
> **Q8:** **Why didn’t you compare SpikeBERT against DistilBERT?**
>
> A8: Due to page limitations, we have included “energy reduction compared to other BERT variants” in the **Appendix E** of our original manuscript. We compare two classic BERT variants, DistilBERT[9] and TinyBERT[10], with our SpikeBERT. We want to state that spiking neural networks and model compressing are two different technological pathways to achieve energy efficiency. Future advancements in neuromorphic hardware are expected to decrease energy consumption further.
>
> **Q9: Table 2: Why is the number of FLOPs consistently larger for SpikeBERT than for FT BERT although SpikeBERT’s energy consumption is much lower? How do you explain that?**
>
> A9: We apologize that “FLOPs/SOPs(G)” term in Table 2 is misleading.  For SNNs, we report their SOPs only, while for ANNs, we report their FLOPs only. We can not tell which model is more energy-efficient by directly comparing absolute value of these two metrics (i.e. FLOPs and SOPs). The FLOPs can be approximately converted into SOPs (See Equation 18 in **Appendix D**):
>
> The number of synaptic operations at the layer $l$ of an SNN is estimated as $SOPs(ξ) = T × γ × FLOPs(l)$, where $T$ is the number of times step required in the simulation, $γ$ is the firing rate of input spike train of the layer $l$.
>
> Appendix D titled “Theoretical Energy Consumption Calculation” explains the relation of FLOPs and SOPs, and also give detailed formulas of energy estimation based on 45nm neuromorphic hardware[6].
>
> **Q10: Table 3: Did you employ the same data augmentation strategies to all methods that you compared SpikeBERT against?**
>
> A10: In Table 3, all reported values, except those shown in “w/o Stage 2” row and “Stage 2 w/o DA” row, employ the same data augmentation strategies for the downstream tasks. In addition, according to our ablation study, data augmentation is not the most important factor (only bring 0.76% performance increase). We think our proposed two-stage knowledge distillation with four types of losses is crucial.
>
>  **Q11: The math and notation in Section 3 are a bit sloppy and not very precise.**
>
> A11: We have carefully checked our formulas in Section 3, and update some notations in our revised manuscripts.
>
> [1] Zhou Z, Zhu Y, He C, et al. Spikformer: When Spiking Neural Network Meets Transformer[C]//The Eleventh International Conference on Learning Representations. 2022.
>
> [2] Lv C, Xu J, Zheng X. Spiking Convolutional Neural Networks for Text Classification[C]. // The Eleventh International Conference on Learning Representations. 2022.
>
> [3] Dosovitskiy A, Beyer L, Kolesnikov A, et al. An Image is Worth 16x16 Words: Transformers for Image Recognition at Scale[C]//International Conference on Learning Representations. 2020.
>
> [4] Yao M, Hu J K, Zhou Z, et al. Spike-driven Transformer[C]//Thirty-seventh Conference on Neural Information Processing Systems. 2023.
>
> [5] Zhou C, Yu L, Zhou Z, et al. Spikingformer: Spike-driven Residual Learning for Transformer-based Spiking Neural Network[J]. arXiv preprint arXiv:2304.11954, 2023.
>
> [6] Horowitz M. 1.1 computing's energy problem (and what we can do about it)[C]. // 2014 IEEE international solid-state circuits conference digest of technical papers (ISSCC). IEEE, 2014: 10-14.
>
> [7] Qiu X, Sun T, Xu Y, et al. Pre-trained models for natural language processing: A survey[J]. Science China Technological Sciences, 2020, 63(10): 1872-1897.
>
> [8] Liu Y, Ott M, Goyal N, et al. Roberta: A robustly optimized bert pretraining approach[J]. arXiv preprint arXiv:1907.11692, 2019.
>
> [9]Tang R, Lu Y, Liu L, et al. Distilling task-specific knowledge from bert into simple neural networks[J]. arXiv preprint arXiv:1903.12136, 2019.
>
> [10]Jiao X, Yin Y, Shang L, et al. Tinybert: Distilling bert for natural language understanding[J]. arXiv preprint arXiv:1909.10351, 2019.
>
>
>
> Finally, we thank you for your promising insights on our paper. We understand that reviewers are not required to read the Appendix section. However, due to the constraint on page limitation, we have written some parts (such as "Discussion of Limitations", "Comparison of DistilBERT/TinyBERT", and "Performance on GLUE") in the Appendix of our original manuscript. Meanwhile, the Appendix.pdf file can be found in our supplementary material. We sincerely hope the Appendix and the Q&A above can address your concerns. If you have any further questions, please let us know~ Thank you very much :)

---

> ### Comment · Area_Chair_Eiv1 · 2023-11-15
> **Comment on authors' rebuttal?**
>
> @Reviewer sLn7: Does the reply by the authors address the issues raised by you? Do you have any follow-up questions or comments?

---

> ### Comment · Reviewer_sLn7 · 2023-11-16
> **Reviewer's response to authors' rebuttal**
>
> I have read the rebuttal and I thank the authors for their thorough response! However, I will not change my rating. I can see this manuscript being an interesting contribution to a workshop on SNNs or a “smaller” NLP conference like NACL or EACL, but I don't feel comfortable accepting it to ICLR. The contribution is too minor and I am not sure why anyone would use their method over variants of DistillBERT which is a method from a few years ago. I am also not sure about the benefits for the broader ICLR community.

---

> > ### Author Response · Authors · 2023-11-23
> > **Response to Reviewer sLn7's concerns on accepting our paper to ICLR**
> >
> > Dear Reviewer sLn7,
> >
> > Thank you for your thoughtful review and valuable feedback on our paper.
> > We appreciate the opportunity to address your concerns and provide additional clarity on our work.
> >
> > While we understand your consideration regarding the venue for our paper and the suggestion to submit it to conferences such as NAACL or EACL, we would like to emphasize that the primary focus of our research lies in advancing the processing of natural languages through a biologically-plausible computing paradigm (i.e., spiking neural networks, SNNs).
> > Our objective is to explore the feasibility of achieving energy-efficient artificial intelligence by SNNs, particularly in the realm of natural language processing.
> > We believe that our study holds significant implications for **brain-inspired computing** and **its applications**.
> >
> > Regarding the utility of our proposed SpikeBERT, we share your concern and acknowledge the importance of ensuring its practical application.
> > We believe that spiking neural networks could become a viable choice for implementing modern AI if their performance is comparable to traditional deep neural networks, while retaining the advantages of lower energy consumption and faster inference, particularly when deployed on neuromorphic hardware.
> > Our ongoing research, including the development of SpikeBERT, is part of a broader initiative aimed at demonstrating the potential of spiking neural networks and facilitating their adoption in mainstream AI applications.
> >
> > We hope this response addresses your concerns and provides a clearer perspective on the motivations and aspirations of our study.
> > If you find our clarifications satisfactory, we kindly ask for your consideration in adjusting your scores accordingly.
> > Your insights are valuable to us, and we appreciate your time and attention to our work.
> >
> > Thank you very much in advance for your consideration.
> >
> > Best regards,
> >
> > The Authors

---

> ### Author Response · Authors · 2023-11-16
> **Further Discussion on the Response of Reviewer sLn7**
>
> Dear Reviewer sLn7,
>
> We acknowledge the insightful comments provided by you and appreciate the opportunity to address your concerns.
>
> **1. What is spiking neural network? Why is it important?**
>
> Spiking neural network (SNN) is a brain-inspired neural network proposed by [1] in 1997, which has been seen as the third generation of neural network models and an important **application of neuroscience**. SNNs use discrete spike trains (0 and 1 only) instead of floating-point values to compute and transmit information, which are quite suitable for implementation on neuromorphic hardware. Therefore, compared to traditional artificial neural networks that run on GPUs, the SNNs offer an energy-efficient computing paradigm to deal with large volumes of data using spike trains for information representation when inference. To some degree, we can regard SNN as **a simulation of neuromorphic hardware** used to handle a downstream deep learning task. Nowadays, the neuromorphic hardware mainly refers to brain-like chips, such as 14nm Loihi2 (Intel), 28nm TrueNorth (IBM), etc. As for the training of SNNs, there are no mature on-chip training solutions currently so the training has to be done on GPUs. However, once SNNs are well trained on GPUs, they can be deployed on neuromorphic hardware for energy-efficient computing (0-1 computing only).
>
> **2. The motivation and contribution of this study.**
>
> Motivation:
>
> There are few works that have demonstrated the effectiveness of spiking neural networks in natural language processing tasks. Current state-of-the-art model is SNN-TextCNN[2], which is based on a simple backbone TextCNN.
>
> However, LLMs like ChatGPT perform very well in many language tasks nowadays, but one of their potential problems is the huge energy consumption (even when inference with GPUs). We want to implement spiking versions of LLMs running on low-energy neuromorphic hardware so that models are able to do **GPUs-free inference** and the **energy consumption** when inference can be significantly reduced. Our proposed SpikeBERT can be seen as the first step. We hope SpikeBERT can lead to future energy-efficient implementations of **large language models on brain-inspired neuromorphic hardware**!
>
> Contributions:
>
> (1) We propose the **first** Transfomer-like SNN for **language** tasks with the same scale as the scale of BERT. Experiments show that SpikeBERT outperforms all SNN baselines in all benchmarks and achieve comparable performance of BERT with much less energy consumption. Note that the inference of SpikeBERT is **GPUs-free** and **energy-efficient**.
>
> (2) Directly-trained SpikeBERT suffers from the problem of gradient vanishing or exploding due to “self-accumulating dynamics”. Therefore, we choose knowledge distillation to train SpikeBERT. We would like to clarify that our approach involves the challenging task of distilling knowledge from BERT into spiking neural networks (SNNs), where the fundamental distinction lies in the use of discrete spikes for computation and information transmission in the student model (SNN), as opposed to the continuous values in the teacher model (BERT). To address this disparity, we introduce a novel two-stage "pre-training + task-specific" knowledge distillation (KD) method. This method incorporates proper normalization across both timesteps and training instances within a batch, enabling a meaningful comparison and alignment of feature representations between the teacher and student models.
>
> **3.** **The performance and baselines concerns.**
>
> SNNs still lag behind ANNs in terms of accuracy yet. Through intensive research on SNNs in recent years, the performance gap between deep neural networks (DNNs) and SNNs is constantly narrowing. SNNs cannot currently outperform DNNs on the datasets that were created to train and evaluate conventional DNNs (they use continuous values). Such data should be converted into spike trains before it can be feed into SNNs, and this conversion might cause loss of information and result in a reduction in performance. Therefore, the comparison is indirect and unfair. In our study, we have conscientiously chosen existing SNNs as baselines for evaluation to provide **a fair and relevant benchmark**. As shown in Table 1, the performance of SpikeBERT surpass all baselines (SNN-TextCNN and Directly-trained Spikformer) in all datasets. This comparison underscores the reasonably good performance of our model.
>
> [1]Maass W. Networks of spiking neurons: the third generation of neural network models[J]. Neural networks, 1997, 10(9): 1659-1671.
>
> [2]Lv C, Xu J, Zheng X. Spiking Convolutional Neural Networks for Text Classification[C]. // The Eleventh International Conference on Learning Representations. 2022.
>
> If you have any further concerns, please let us know and we will do our best to address your concerns! Thank you very much :-D

---

> ### Author Response · Authors · 2023-11-21
> **To Reviewer sLn7**
>
> Dear Reviewer sLn7,
>
> We sincerely appreciate your thorough review and the valuable comments you provided for our paper.
> We have carefully considered each point and have addressed them in detail in our rebuttal.
> Furthermore, we have implemented several improvements in our revised manuscript based on all reviewers' suggestions.
> Specially, according to your suggestions, (1) we have reported standard deviations and bolded the numbers of fine-tuned BERT in Table 1; (2) we have added our research motivations in Appendix H, including what is spiking neural network, why is it important, and our motivations; (3) we have updated the mathematical notations in Section 3.
>
> As the Author-Review Discussion period is drawing to a close with only two days remaining, we would like to ensure that all your concerns have been adequately addressed.  If there are any questions or unresolved issues, we are eager to provide further clarification or make necessary revisions.
>
> Best regards,
>
> The Authors

---

> ### Author Response · Authors · 2023-11-23
> **Response to Reviewer sLn7's concerns on our paper's benefits for the broader ICLR community**
>
> Dear Reviewer sLn7,
>
> Thank you very much for taking the time to carefully review my submission and providing valuable feedback.
> We understand your uncertainty regarding the broader benefits of our paper on the intersection of Spiking Neural Networks (SNN) and Natural Language Processing (NLP) for the ICLR community.
> We would like to clarify the value and contributions of our research to the broader ICLR community through the following points:
>
> **1. Innovative Interdisciplinary Fusion**
>
> Our study focuses on brain-inspired computing (especially on SNNs), a highly relevant and cutting-edge research area.
> Actually,  the track of our paper is **"applications to neuroscience & cognitive science"**.
> By delving into the intersection of **neuroscience** and **computer science**, we aim to introduce a biologically-plausible and more effective paradigm for information processing to the ICLR community, which may pave the way for more sustainable and efficient AI solutions.
> We would like to emphasize that, our ongoing research, including the development of SpikeBERT, is part of a broader initiative aimed at demonstrating the potential of spiking neural networks and facilitating their adoption in mainstream AI applications.
>
> **2. Potential Advantages for Other Reseach Topics**
>
> The combination of SNNs and NLP may offer potential advantages in handling temporal information, simulating neural activity, and addressing practical issues of interest to the ICLR community.
> For example, in our responese Q6 to Reviewer is3e, we mention that how to utilize SNNs to process **time series tasks** may be a good reseach topic.
>
> We hope these points clarify the benefits of our research to the ICLR community.
> If our explanations meet your expectations, we kindly request you to reflect these clarifications in your scoring.
> Your feedback is highly valued, and we sincerely appreciate the time and attention you have dedicated to reviewing our work.
>
> Best regards,
>
> The Authors

---

> > ### Comment · Reviewer_sLn7 · 2023-11-23
> >
> > I am sorry but I think you just have to accept my decision and stop sending me response after response.

---

### Official Review · Reviewer_17Yy · 2023-10-27

**Soundness:** 3 good
**Presentation:** 3 good
**Contribution:** 3 good
**Rating:** 8
**Confidence:** 5

**Summary:**

The paper proposes SpikeBERT, a spiking BERT model designed for language tasks based, and describes a two-stage distillation method employed in its training.

The authors conduct experiments on several text classification tasks on English and Chinese datasets. The results show that SpikeBERT outperforms state-of-the-art spiking neural networks and achieves comparable results to BERT on text classification tasks for English and Chinese, while consuming significantly less energy.

**Strengths:**

1. The authors provide the necessary background on spiking neural networks (SNNs) and Spikformer architecture.

2. This work is the first Transformer-based SNNs for language tasks, and achieve state-of-the-art performance on text classification tasks.

3. The authors present an ablation analysis for all their contributions, and compare SpikeBERT with other BERT variants like TinyBERT and DistilBERT on Appendix.

**Weaknesses:**

1. Although SNNs can reduce the energy consumption when inference, the proposed two-stage distillation method may lead to more energy costs when training. Can you explain this matter?

2. In Figure 3(b), it seems there are no emergent abilities in the SpikeBERT, which is different from non-spiking large language models.

**Questions:**

Q: I wonder why the authors choose BERT as their teacher model.

If the authors can respond reasonably to all my questions and comments, I will improve the score of this manuscript.

---

> ### Author Response · Authors · 2023-11-13
>
> Thank you for your valuable comments!
>
> **Q1: The proposed two-stage distillation method may lead to more energy costs when training. Can you explain this?**
>
> A1: For spiking neural networks, the energy consumption is mainly reduced at the inference time. Once spiking neural networks (SNNs) are well software-trained, they can be deployed on neuromorphic hardware (such as 45nm neuromorphic hardware[1]) for energy-efficient computing (binary-value computing). However, mature on-chip training solutions are not yet available, and it remains a great challenge due to the lack of efficient training algorithms, even in a software training environment. Thank you for pointing it out.
>
> **Q2:** **It seems** **there are no emergent abilities in the SpikeBERT.**
>
> A2: Firstly, the emergent ability of large language models was proposed by [2]. Large language models usually refer to large **generative** language models, which are mostly decoder-only and mainly used for text generation, such as ChatGPT. SpikeBERT is an encoder-only language **representation** model for language understanding tasks, which is similar to BERT. However, SpikeBERT can be easily extended to the decoder-only models because they are all Transformer-based.
>
> Secondly, as discussed in Section 4.5, the reason why the accuracy of SpikeBERT is generally insensitive to the model depths is that the gradients error may accumulate with the increase of model depths due to the surrogate gradients. What’s more, the depths we used on the experiments were only 8, 12 and 18. Should you wish to explore the potential performance fluctuations across a broader range of depths, we are readily prepared to conduct additional experiments at your behest.
>
> **Q3:** **Why the authors choose BERT as their teacher model?**
>
> A3: Our approach is applicable to any model similar to BERT or RoBERTa. Existing literature[3] indicates that BERT and RoBERTa exhibit minimal substantive differences in downstream tasks, and both models share a similar network architecture. The divergence between these models is primarily observed in the aspect of masking strategy, input tokenization and training strategy [4], yet these differences do not impact the conclusions drawn in this paper. What’s more, as BERT serves as a representative example of large pre-trained models, we chose BERT as the teacher model.
>
>
>
> [1] Horowitz M. 1.1 computing's energy problem (and what we can do about it)[C]. // 2014 IEEE international solid-state circuits conference digest of technical papers (ISSCC). IEEE, 2014: 10-14.
>
> [2] Wei J, Tay Y, Bommasani R, et al. Emergent abilities of large language models[J]. arXiv preprint arXiv:2206.07682, 2022.
>
> [3] Qiu X, Sun T, Xu Y, et al. Pre-trained models for natural language processing: A survey[J]. Science China Technological Sciences, 2020, 63(10): 1872-1897.
>
> [4] Liu Y, Ott M, Goyal N, et al. Roberta: A robustly optimized bert pretraining approach[J]. arXiv preprint arXiv:1907.11692, 2019.

---

> > ### Comment · Reviewer_17Yy · 2023-11-13
> >
> > Thx!  I believe the author has addressed all my concerns very well. I think it's a high-quality work and deserves to be accepted.

---

> > > ### Author Response · Authors · 2023-11-23
> > > **To Reviewer 17Yy**
> > >
> > > Dear Reviewer 17Yy,
> > >
> > > We are pleased that the concerns raised by you have been successfully addressed.
> > > We would like to reiterate our deep appreciation for your dedicated time and effort in scrutinizing our paper and providing valuable feedback.
> > >
> > > Best regards,
> > >
> > > The Authors

---

### Official Review · Reviewer_is3e · 2023-10-31

**Soundness:** 3 good
**Presentation:** 3 good
**Contribution:** 4 excellent
**Rating:** 8
**Confidence:** 4

**Summary:**

The authors propose SpikeBERT, a spiking neural network (SNN) architecture for language tasks. SpikeBERT extends and improves Spikformer architecture to process text instead of images. It replaces certain modules in Spikformer to make it suitable for language tasks.
The approach uses a two-stage knowledge distillation method to train SpikeBERT: First stage is pre-training distillation using a large unlabeled corpus to align embeddings and features. Second stage is task-specific distillation using a fine-tuned BERT on a downstream task as teacher. The model is evaluated on 6 English and Chinese text classification datasets: it outperforms prior SNN methods and achieves comparable accuracy to BERT. The estimated theoretical energy consumption is much lower for SpikeBERT as compared to traditional approaches.

**Strengths:**

Advantages:
Uses a highly scalable Transformer-based architecture as the backbone and outperforms prior SNN methods by 3.49% on average across 6 datasets.
Two-stage distillation allows pre-training on large unlabeled data.
Feature alignment loss aligns hidden representations.
Data augmentation further facilitates distillation.
Evaluated on diverse English and Chinese datasets: works well for both English and Chinese text classification.
Significantly reduces theoretical energy consumption (by 27.82% compared to fine-tuned BERT).

The claims are reasonably supported by the results. The proposed SpikeBERT outperforms prior SNN methods significantly and achieves comparable accuracy to BERT on multiple datasets. Ablation studies provide insights into model architecture and training.

**Weaknesses:**

Potential weaknesses include:
The method relies on the teacher ANN, so can not learn directly from the data.
The method does not address zero-shot generalization to novel language tasks, which is the main appeal of the LLMs.
Fails to capture fine-grained word semantics well.
Requires GPUs with large memory due to additional time dimension.
Energy reduction based on theoretical estimates, actual hardware measurements would be more compelling.

**Questions:**

The approach was evaluated on datasets created for ANNs, not neuromorphic data. It would be interesting to consider using e.g. a neuromorphic cochlea for speech signal.
Adding scaling experiments would be helpful - trying bigger versions of SpikeBERT with more layers, heads and timesteps to explore the scaling law.
It would be helpful to provide visualizations of the learned spike patterns to offer insights into model operation and interpretability.
How would the chioce of alternate surrogate gradient functions would impact training convergence and accuracy?

---

> ### Author Response · Authors · 2023-11-13
>
> Thank you for your valuable comments!
>
> **Q1:** **The method relies on the teacher ANN, so can not learn directly from the data.**
>
> A1: As a matter of fact, we have tried to directly train our SpikeBERT on downstream tasks, and have reported its performance in Table1 (“Directly-trained Spikformer” row). However, as discussed in Section 1, **deep** spiking neural networks (SNNs) directly trained with backpropagation through time using surrogate gradients will suffer from the problem of gradient vanishing or exploding due to “self-accumulating dynamics”.
>
> Previous SNN work[1] on natural language processing used a simple network TextCNN as their backbone. However, our SpikeBERT is the first Transformer-liked SNN for language tasks, whose scale is the same as the scale of BERT. Therefore, we choose to use knowledge distillation for training our SpikeBERT so that the deviation of surrogate gradients in spiking neural networks will not be rapidly accumulated.
>
> **Q2:** **The method does not address zero-shot generalization to novel language tasks, which is the main appeal of the LLMs.**
>
> A2: Zero-shot capability refers to the generation capability of language **generative** model, which are mostly decoder-only and mainly used for text generation, such as GPT3 and ChatGPT. Our SpikeBERT is an encoder-only spiking language **representation** model for language understanding tasks, which is similar to BERT. However, SpikeBERT can **be easily extended** to the decoder-only models because they are all Transformer-based. Now we are focusing on how to implement spiking version of GPT-liked language model, which may lead to future energy-efficient implementations of large language models.
>
> **Q3:** **SpikeBERT fails to capture fine-grained word semantics well.**
>
> A3: The reviewer may concern the ability of SpikeBERT to capture fine-grained word semantics due to the spiking property. Inspired by your comments, we design a set of experiments to show how well SpikeBERT can capture word semantics. For Transformer-liked models, the self-attention module is the key for capturing fine-grained word semantics. To prove our SpikeBERT has successfully captured the word semantics, we will conduct a visualization experiment on **attention map** in spiking-self-attention(SSA) module (See Q8).
>
> **Q4: SpikeBERT requires GPUs with large memory due to additional time dimension.**
>
> A4: Due to the additional time dimension T, SpikeBERT indeed requires large memory on GPUs when **training**. However, once spiking neural networks (SNNs) are well software-trained, they can be deployed on neuromorphic hardware (such as 45nm neuromorphic hardware[2]) for energy-efficient **inference** (binary-value computing), which is GPU-free.
>
> **Q5: Energy reduction based on theoretical estimates, actual hardware measurements would be more compelling.**
>
> A5: Our theoretical energy consumption calculation in Appendix D is based on a classic 45 nm brain-inspired neuromorphic hardware[2], which has been mass-produced. Therefore, theoretical estimates of many previous works like SNN-TextCNN[1] and Spikformer[3] are all based on this chip. We follow them to prove our SpikeBERT is energy-efficient.
>
> **Q6:** **It would be interesting to consider using e.g. a neuromorphic cochlea for speech signal.**
>
> A6: This is a very promising insight! And after careful consideration, we think that speech data is born with time step dimension T, which may be very suitable for SNNs! However, after conducting a meticulous literature review these days, we find current works[4][5][6] on “SNNs + time series tasks” are not that promising (a little trivial). We think how to process time series data by SNNs remains to be researched. Thank you!

---

> ### Author Response · Authors · 2023-11-13
>
> **Q7:** **Adding scaling experiments would be helpful.**
>
> A7: It is a good suggestion. In Fig 3(a)(b), we conducted preliminary experiments to increase the time step T and the number of model layers. But the conclusion shows that with the increase of these two hyper-parameters, the performance gradually stops increasing. We are willing to add more hyper-parameter experiments, but model training may take too long time (we can’t finish all these experiments within 2 weeks), so we may add these results in our camera-ready version upon acceptance. We sincerely hope you can understand us on this matter. Thank you!
>
> **Q8:** **It would be helpful to provide visualizations of the learned spike patterns to offer insights into model operation and interpretability.**
>
> A8: It is a good suggestion. We will visualize the **attention map** in spiking-self-attention(SSA) module. We will report the experiment results and the corresponding visualizations in our revised manuscript once the experiment is completed. Thanks for pointing it out :)
>
> **Q9: How would the chioce of alternate surrogate gradient functions would impact training convergence and accuracy?**
>
> A9: We have followed previous SNN works[1][3][7] to choose the widely-used Arctangent-like surrogate gradient function. It may take so much time to train SpikeBERTs with different surrogate gradient functions that we can not complete all experiments before the rebuttal deadline. We may report these results in our camera-ready version upon acceptance.
>
>
> [1] Lv C, Xu J, Zheng X. Spiking Convolutional Neural Networks for Text Classification[C]. // The Eleventh International Conference on Learning Representations. 2022.
>
> [2] Horowitz M. 1.1 computing's energy problem (and what we can do about it)[C]. // 2014 IEEE international solid-state circuits conference digest of technical papers (ISSCC). IEEE, 2014: 10-14.
>
> [3] Zhou Z, Zhu Y, He C, et al. Spikformer: When Spiking Neural Network Meets Transformer[C]//The Eleventh International Conference on Learning Representations. 2022.
>
> [4] Fang H, Shrestha A, Qiu Q. Multivariate time series classification using spiking neural networks[C]//2020 International Joint Conference on Neural Networks (IJCNN). IEEE, 2020: 1-7.
>
> [5] Sharma V, Srinivasan D. A spiking neural network based on temporal encoding for electricity price time series forecasting in deregulated markets[C]//The 2010 international joint conference on neural networks (IJCNN). IEEE, 2010: 1-8.
>
> [6] Gaurav R, Stewart T C, Yi Y. Reservoir based spiking models for univariate Time Series Classification[J]. Frontiers in Computational Neuroscience, 2023, 17: 1148284.
>
> [7] Fang W, Yu Z, Chen Y, et al. Incorporating learnable membrane time constant to enhance learning of spiking neural networks[C]//Proceedings of the IEEE/CVF international conference on computer vision. 2021: 2661-2671.

---

### Author Response · Authors · 2023-11-18
**To All Reviewers**

We express our gratitude to all the reviewers for their valuable insights, which have significantly enhanced the quality of our manuscript. In response to the constructive feedback, we have implemented several improvements:

(1) **Visualization of Attention Maps**: We have included visualizations of attention maps generated by the spiking-self-attention (SSA) module of SpikeBERT in the Appendix. This addition illustrates SpikeBERT's capability to capture fine-grained semantics at the word level.

(2) **Reporting Standard Deviations**: In Table 1, we now present the standard deviations of accuracy alongside their means. The most noteworthy results are highlighted using bold fonts.

(3) **Clarity in Mathematical Notations**: To enhance clarity, we have updated the mathematical notations in Section 3.

(4) **Addressing Research Motivation**: The motivation behind our study has been elaborated in the Appendix.

(5) **Additional Experiments with SpikeBERT**: While we endeavor to conduct additional experiments on SpikeBERT with varying time steps, different model depths, and alternative surrogate gradient functions, the completion before the rebuttal deadline might not be feasible. These results will be incorporated into the revised version upon acceptance.

(6) **Comprehensive Reviewer Feedback**: All comments from the reviewers have been addressed in the revised version.

(7) **Thorough Revision**: We have revised the paper thoroughly and carefully.

We sincerely appreciate the reviewers' dedication to the peer-review process and are confident that these enhancements contribute significantly to the overall strength and clarity of the paper.

---

### Public Comment · ~Xingu_Meng1 · 2023-11-25

I would like to thank the authors for a nice piece of work! There seems to be some innovation related to KD to improve SNN performance on language tasks. However, I would like to point out a few critical issues (in my opinion) with the paper and the authors' response. Should this paper be accepted, it would be great if the authors can correct these mistakes so as not to mislead the readers.

1. The authors mentioned in their multiple responses that [1] is a classic 45 nm brain-inspired neuromorphic hardware, which has been mass-produced. This is totally incorrect, as [1] is a neither neuromorphic hardware, nor it is mass-produced. It is one of the seminal papers by Prof. Mark Horowitz from Stanford University that showed nice scaling trends of computing performance and energy for Von-Neumann architectures. Most SNN papers refer to this paper (quite rightly so) for the energy numbers of multiplication and addition operations.

2. The authors mentioned they used the compute energy model from [1] since earlier CNN-based SNN papers did the same. However, since CNNs are compute-heavy, this model might be fair to use. However, BERT and other language models are known to be memory-heavy, and hence, only considering compute might not reflect the true energy consumption. I think the authors really need to include the memory energy in Eq. (19) for a fair evaluation.

3. It seems that the authors missed comparing their results with SpikingGPT [2], which is a recently proposed SNN model trained for language tasks. SpikeGPT was released to Arxiv on February 27, 2023, which is almost 7 months before the ICLR deadline. The authors cite spikeGPT in the introduction, but (most probably) not in a correct way. They mention that the backbone network employed in spikeGPT is relatively simplistic, which significantly lowers the upper bound on the performance. I do not think such a remark is appropriate since spikeGPT develops models which have more number of parameters (216M vs 109M) compared to what is reported in this paper. Note that I am neither an author nor affiliated to the SpikeGPT work in any way.

By no means, I intend to attack the value proposition of this paper with my comments. The paper may be accepted by its own merits, and since I am not an official reviewer, I would not give any comments on the scientific merits of this paper. However, the issues I mentioned above should be resolved to avoid critical misinterpretations within the readership and the community.

[1] Horowitz M. 1.1 computing's energy problem (and what we can do about it), IEEE international solid-state circuits conference digest of technical papers (ISSCC). IEEE, 2014: 10-14.

[2] https://arxiv.org/abs/2302.13939

---

> ### Public Comment · ~Rui-Jie_Zhu1 · 2023-11-28
> **Comment from SpikeGPT author**
>
> As the author of SpikeGPT, I would like to extend my gratitude to the commenter for bringing up such an insightful question and for mentioning my work!  I also would like to add some context to this discussion:
>
> 1. It is indeed noteworthy that SpikeGPT has a larger parameter count (216M) compared to the model discussed in this paper. While it's true that our model possesses more parameters and generative capabilities, it is also common for BERT models to have fewer parameters than GPT models in general.
> 2. Although SpikeGPT is primarily designed for Natural Language Generation (NLG), we have also explored its performance on Natural Language Understanding (NLU) tasks. Our NLU experiments are preliminary and serve as a baseline, while the GLUE benchmark used by SpikeBERT in this paper is a more authoritative and comprehensive measure for evaluating NLU tasks.
>
> I appreciate the opportunity to discuss these nuances and thank everyone involved for such a constructive and engaging dialogue~ Thanks!

---

> ### Public Comment · ~Jiache_lu1 · 2023-11-29
>
> Dr. Xingu Meng,
>
> As a researcher studying on neuromorphic chips, I am also focusing on this paper.
>
> I would like to point out that you may misunderstand the Eq. (19), which presents the energy consumption estimation when inference, rather than when training. I guess the memory energy you mentioned is mainly about the training step, which relies on GPU memory. In previous studies like Spikformer (ICLR 2023) or Spike-driven Transformer (NeurIPS 2023), the energy consumptions they reported in their papers are all during inference, which is memory-free.
>
> SpikeBERT follows the architecture of Spikformer and you can find Spikformer’s energy formula in Appendix C.2 of [1], which is similar to SpikeBERT’s energy formula. The energy numbers of multiplication and addition operations ($E_{AC}$ and $E_{MAC}$) follow [2]. Different from SpikeBERT, the 45 nm hardware cited in Spikformer is [3]. You can find that in [3], the authors state that (1) the number of $E_{AC}$ and $E_{MAC}$ is also from [2]. (2) “In the encoder layer of vanilla SNNs (n = 1), FLOPs are MAC operations that are the same as CNNs, because the work of this layer is to transform analog inputs into spikes. In addition, all other Conv and FC layers transfer spikes and execute AC operations to accumulate weights of postsynaptic neurons.” Therefore, I think: (1) the authors of SpikeBERT should cite [3] instead of citing [2], i.e., modify it as “45nm hardware technology[3]”. (2) both CNN-based and MLP-based SNNs can utilize this numbers to calculate energy consumption when inference.
>
> But one of my concerns on this paper is about the energy estimation of the word embedding layer in SpikeBERT.
> It seems that the authors treat the word embedding layer as a MAC (multiply-and-accumulate) operation according to Eq. (19).
> I think it may be not appropriate, because word embedding layer, as far as I see, is just for key-value mapping, which does no dynamic operations like MAC.
> From the perspective of chips designing, the energy consumption of word embedding layer includes 2 parts: the key-value maintaining energy and the mapping energy. Maintaining energy depends on static random-access memory (SRAM) in chips and how many bits keys and values take.
> Mapping energy is mainly for searching and reading the value from the memory address of key, which consume much less energy than MAC operations.
> In general, the dynamic operations consume much more energy than static operations in neuromorphic chips.
> Therefore, I think the authors may even overestimate (not sure, depending on chips) the inference energy consumption.
>
> Note that this is not an academic attack on you or the authors, just a discussion. I am quite pleased to discuss on this interesting topic. So, if there are some mistakas in my comment, it is okay to point it out.
>
> [1] Spikformer: When Spiking Neural Network Meets Transformer. The Eleventh International Conference on Learning Representations (ICLR). 2023.
>
> [2] Horowitz M. 1.1 computing's energy problem (and what we can do about it), IEEE international solid-state circuits conference digest of technical papers (ISSCC). IEEE, 2014: 10-14.
>
> [3] Attention spiking neural networks. IEEE Transactions on Pattern Analysis & Machine Intelligence (TPAMI). IEEE, 2023: 1-18.

---

> ### Public Comment · ~Yongqi_Leng1 · 2023-12-05
>
> To Xingu Meng,
>
> As a fellow researcher in the field of spiking neural networks and natural language processing, I share your interest in the SpikeBERT proposed by the authors.
>
> I have also read SpikeGPT in arxiv a few months ago. Upon a careful examination of SpikeGPT, I tend to categorize it as more of an ANN-SNN hybrid network rather than a pure SNN. The reason is that SpikeGPT contains both floating-point computing and spike computing. This perspective aligns with the observations made by a SpikeGPT’s reviewer in the OpenReview ICLR 2024 (https://openreview.net/forum?id=ShOT80BjUZ), who noted “SpikeGPT introduces some float-point operations including float-point multiplication, division, and exponentiation. This makes SpikeGPT different from traditional spiking neural networks”. You can also check this point from SpikeGPT’s source code on Github (https://github.com/ridgerchu/SpikeGPT).
>
> In contrast, my investigation into the supplementary material accompanying SpikeBERT revealed a strict adherence to the operation rules of spiking neural networks, exclusively employing spike computing, so I tend to treat SpikeBERT as a pure SNN. Therefore, I think it may be unfair to compare results with SpikeGPT and SpikeBERT directly, for their divergent operation modes. But I concur with your observation that citing SpikeGPT in the context of the backbone issue may not be appropriate.

---

### Meta-Review · Area_Chair_Eiv1 · 2023-12-14

**Metareview:**

This submission explores the transfer of knowledge from the transformer-based BERT model to the spiking neuron-based architectures which require much lower energy consumption. They propose the SpikBERT model, which is based on prior work of Spikformer, and use a two-stage pretraining + task-specific knowledge distillation process to train the spiking neuron model.They show that their model achieves higher accuracy than prior work using spiking neurons.

Weaknesses:
a) High degree of similarity to the Spikformer architecture, with only a few modifications actually needed to convert the model from the vision to the textual domain - 1) patch splitting module replaced with a word embedding layer, 2) reshaping of the attention map, and 3) replacing convolutions with linear layers.
b) The proposed knowledge distillation is also not novel, as published work on ANN to SNN knowledge distillation exists, including some which have not been cited, e.g. Constructing Deep Spiking Neural Networks from Artificial Neural Networks
with Knowledge Distillation, CVPR 2023.
Most importantly, the distillation method used in this work is borrowed from a previously withdrawn ICLR 2023 submission. Note that Eq 6) in the submission is the same as Eq 8) in the cited work "Self-Architectural Knowledge Distillation for Spiking Neural Networks" (https://openreview.net/forum?id=QwFw-CcUb10), accounting for BN and Conv replaced with LN and MLP. This cited work was withdrawn from ICLR 2023, with unresolved issues as pointed out by the reviews visible in the link.
c) Further, as noted by some reviewers, the writing can indeed be improved.

Weaknesses a) and b) negate both claimed contributions of the submission; it is unclear what the remaining key contributions or novelties of the submission are. in its current form, the submission does not provide any additional insight into the task or any new schemes to better train SNNs.

The submission could have been strengthened in many ways including but not limited to - exploring, comparing, and improving distillation (comparison of per-layer v/s last-layer feature matching, or new distillation techniques, etc), or analyzing the quality and possible benefits/pitfalls of the distilled SNN when compared to the original ANN (are there specific failure modes)? The current manuscript feels more like "we tried this and these are the results we got", without additional insight or learnings.

The ACs would like to note that the direction of the work is indeed interesting and deserves a place at ICLR. However, the current submission does not clear the bar for acceptance owing to the lack of novelty and lack of insights and contributions.

**Justification For Why Not Higher Score:**

Weaknesses listed above negate all claimed contributions. It is unclear what this submission actually provides in terms of novelty and learning.

**Justification For Why Not Lower Score:**

N/A

---

### Decision · Program_Chairs · 2024-01-16

Reject